# Fgf-signaling is compartmentalized within the mesenchyme and controls proliferation during salamander limb development

Sruthi Purushothaman[1], Ahmed Elewa[2], Ashley W Seifert[1]*

[1]Department of Biology, University of Kentucky, Lexington, United States; [2]Cold Spring Harbor Laboratory, Cold Spring Harbor, United States

**Abstract** Although decades of studies have produced a generalized model for tetrapod limb development, urodeles deviate from anurans and amniotes in at least two key respects: their limbs exhibit preaxial skeletal differentiation and do not develop an apical ectodermal ridge (AER). Here, we investigated how *Sonic hedgehog* (*Shh*) and *Fibroblast growth factor* (*Fgf*) signaling regulate limb development in the axolotl. We found that *Shh*-expressing cells contributed to the most posterior digit, and that inhibiting Shh-signaling inhibited *Fgf8* expression, anteroposterior patterning, and distal cell proliferation. In addition to lack of a morphological AER, we found that salamander limbs also lack a molecular AER. We found that amniote and anuran AER-specific *Fgfs* and their cognate receptors were expressed entirely in the mesenchyme. Broad inhibition of Fgf-signaling demonstrated that this pathway regulates cell proliferation across all three limb axes, in contrast to anurans and amniotes where Fgf-signaling regulates cell survival and proximodistal patterning.

*For correspondence:
awseifert@uky.edu

Competing interests: The authors declare that no competing interests exist.

## Introduction

Limb development is an ideal model to investigate how cellular and molecular networks exhibit plasticity or resilience during tetrapod evolution. Since the turn of the twentieth century the limb has endured as a fundamental model to investigate morphogenesis, cellular growth, and differentiation during embryonic development. From studies spanning comparative embryology through developmental genetics, limb development has provided deep insight into mechanisms underlying pattern formation, genotype-phenotype relationships, and the complexity of molecular networks (*Swett, 1937*; *Wolpert, 1969*; *Zeller et al., 2009*). Perhaps equally studied as a fully developed structure, the limb has also served as a model for evolutionary biologists seeking to reconstruct tetrapod ancestry and how micro- and macro-evolutionary changes might explain major events in vertebrate evolution such as the fin to limb transition (*Fröbisch and Shubin, 2011*; *Shubin and Alberch, 1986*).

The two most important embryonic models for modern limb development studies have been chicken and mice. Chicken embryos were integral to our understanding of limb bud outgrowth and morphogenesis aided by the availability of genetic limb mutants and the ability to discern skeletal defects in the wing caused by surgical manipulation (*Saunders, 1948*; *Summerbell et al., 1973*). With the advent of stable transgenesis, mice became the model of choice to investigate the molecular basis of limb development. Together, data from chicken, mice and frogs have been synthesized into contemporary models covering limb development in tetrapods (*Zeller et al., 2009*; *Zuniga, 2015*). In these models, all vertebrate limb buds utilize the same molecular network (e.g.

**eLife digest** Salamanders are a group of amphibians that are well-known for their ability to regenerate lost limbs and other body parts. At the turn of the twentieth century, researchers used salamander embryos as models to understand the basic concepts of how limbs develop in other four-limbed animals, including amphibians, mammals and birds, which are collectively known as "tetrapods". However, the salamander's amazing powers of regeneration made it difficult to carry out certain experiments, so researchers switched to using the embryos of other tetrapods – namely chickens and mice – instead.

Studies in chickens, later confirmed in mice and frogs, established that there are two major signaling centers that control how the limbs of tetrapod embryos form and grow: a small group of cells known as the "zone of polarizing activity" within a structure called the "limb bud mesenchyme"; and an overlying, thin ridge of cells called the "apical ectodermal ridge". Both of these centers release potent signaling molecules that act on cells in the limbs. The cells in the zone of polarizing activity produce a molecule often called Sonic hedgehog, or Shh for short. The apical ectodermal ridge produces another group of signals commonly known as fibroblast growth factors, or simply Fgfs.

Several older studies reported that salamander embryos do not have an apical ectodermal ridge suggesting that these amphibian's limbs may form differently to other tetrapods. Yet, contemporary models in developmental biology treated salamander limbs like those of chicks and mice. To address this apparent discrepancy, Purushothaman et al. studied how the forelimbs develop in a salamander known as the axolotl.

The experiments showed that, along with lacking an apical ectodermal ridge, axolotls did not produce fibroblast growth factors normally found in this tissue. Instead, these factors were only found in the limb bud mesenchyme. Purushothaman et al. also found that fibroblast growth factors played a different role in axolotls than previously reported in chick, frog and mouse embryos. On the other hand, the pattern and function of Shh activity in the axolotl limb bud was similar to that previously observed in chicks and mice.

These findings show that not all limbs develop in the same way and open up questions for evolutionary biologists regarding the evolution of limbs. Future studies that examine limb development in other animals that regenerate tissues, such as other amphibians and lungfish, will help answer these questions.

*Shh, Fgfs, Bmps, Wnts* and retinoic acid) governed by *Hox* genes and controlled by two major signaling centers; the zone of polarizing activity (ZPA) and the apical ectodermal ridge (AER).

And then there are the urodeles. With tetrapod monophyly solidly supported by molecular and morphological data (*Ahlberg and Milner, 1994*; *Marjanović and Laurin, 2013*), for scientists investigating the evolution and development of vertebrate limbs, salamanders and newts have longed proved problematic (*Holmgren, 1933*). Although salamander and newt embryos were used to uncover key principles regarding specification of the prospective limb field and establishment of the primary limb axes (i.e., proximal-distal, anterior-posterior and dorsal-ventral) (*Harrison, 1918*; *Stocum and Fallon, 1982*; *Swett, 1937*), it was also observed that urodele limb development deviated from anurans and amniotes in at least two key respects: skeletal specification exhibited preaxial dominance (anterior elements form before posterior elements) rather than postaxial dominance (*Shubin and Alberch, 1986*) and urodele limb buds did not form an apical ectodermal ridge (AER) (*Sturdee and Connock, 1975*; *Tank et al., 1977*). In addition to these important developmental differences, adult urodeles differ from anurans and amniotes in their ability to completely regenerate an amputated limb. With these ideas in mind, we sought to investigate limb development in salamanders to determine whether morphological and molecular data support a unified model of limb development that includes or excludes urodeles.

## Results

### Digit specification and differentiation are uncoupled during axolotl forelimb development

In order to study axolotl limb outgrowth and axis specification, we first staged developing limbs on the basis of external morphology and skeletal chondrification at 20–21°C (*Figure 1A–B* and *Figure 1—figure supplement 1*). As the limb bud emerged from the flank, limb mesenchyme expanded directly above the body wall musculature and at no time during limb development did we observe an AER or thickening of the limb ectoderm (*Figure 1C*). In agreement with previous work in salamanders (*Holmgren, 1933*; *Nye et al., 2003*; *Shubin and Alberch, 1986*), we found that cartilage condensations of the limb skeleton formed proximal to distal, and within the zeugopod and autopod, anterior to posterior (preaxial dominance) (*Figure 1D*). While this was strictly true for the radius and ulna, alcian blue staining always demonstrated digit I and II forming together between stages 47 and 48 (*Figure 1D*). To further examine chondrogenesis in the limb, we identified axolotl *Sox9* (*Supplementary file 1*) and used its expression to analyze condensing mesenchymal cells prior to cartilage formation (*Wright et al., 1995*) (*Figure 1E* and *Figure 1—figure supplement 2*). We first observed *Sox9* expression at stage 45 in a broad proximal area of the axolotl forelimb bud

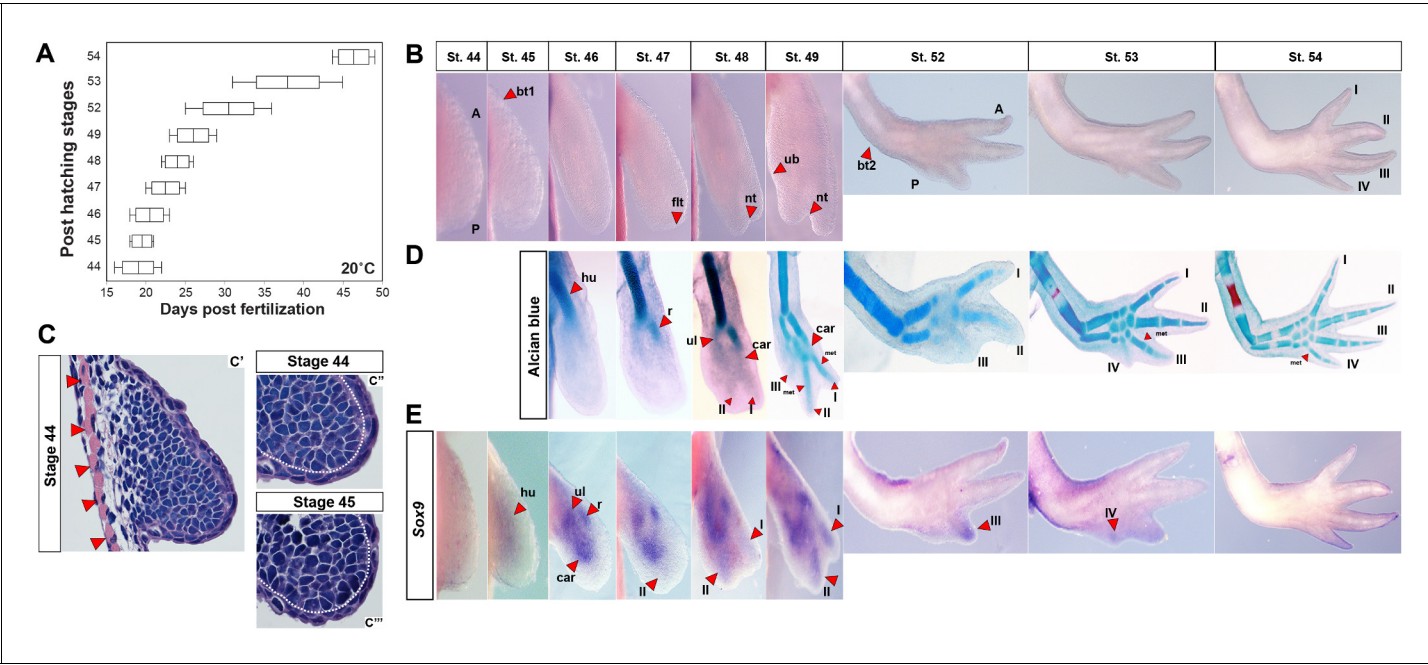

**Figure 1.** Skeletal chondrification proceeds anterior to posterior within the zeugopod and autopod during axolotl forelimb development. (A) Box plot depicting post-hatching limb stages at 20°C (n = 35 total). Central line within the box = median number of larvae for that stage; edges of the box = 25th and 75th quartiles and the whiskers = the interquartile range. (B) Normal, identifiable stages viewed dorsally with anterior (A) at the top and posterior (P) at the bottom (not to scale). For scaled limb stages see *Figure 1—figure supplement 1*. Key stage indicators used: st. 45: bending of limb bud at anterior body wall junction (bt1), st. 46: length doubling along proximodistal axis, st. 47: dorsoventral flattening of distal bud (flt), st. 48: notch (nt) appears separating digit 1 and 2, st. 49: ulnar bulge (ub) appears, st. 52: elbow bend appears (bt2) along with prominent separation of three digits and st. 54: digit four present. (C) H and E stained forelimb buds at stages 44 (C'–C") and 45 (C'") show no evidence of an AER and skeletal muscle (red arrows) underlies the emerging limb bud. White dotted lines (C"–C"') delineate mesenchyme from ectoderm. Ectoderm covering the limb is consistent with flank ectoderm (1–2 cells thick). (D) Alcian blue-Alizarin red staining reveal preaxial pattern of chondrogenesis (n = 3–4/stage). Cartilage condensations abbreviated: humerus (hu), radius (r), ulna (ul), carpal (car), metacarpal (met) and digits 1–4 (I–IV). (E) Expression domains of the pre-condensation marker *Sox9* during forelimb development. *Sox9* is diffusely expressed at stage 45, whereas three discreet domains of *Sox9* expression begin at stage 46 marking the ulna (ul), radius (r) and carpals (car). *Sox9* expression in presumptive digit II is first observable at stage 47 and precedes digit I expression which occurs during late stage 48. *Sox9* expression in the remaining digits occurs anterior to posterior.

The online version of this article includes the following figure supplement(s) for figure 1:

**Figure supplement 1.** Forelimb development stages scaled with respect to final larval limb.
**Figure supplement 2.** Specification of digit II occurs before digit I.

(*Figure 1E*). *Sox9* expression at stage 46 appeared in a centrally located domain corresponding to the future radius, ulna and carpals (*Figure 1D–E*). However, in contrast to the Alcian blue staining pattern, *Sox9* expression in the presumptive digits appeared sequentially II-I-III-IV and this result was consistent across 13/15 (86.66%) limbs analyzed across stage 48 (*Figure 1E* and *Figure 1—figure supplement 2*). At stage 49, *Sox9* expression expanded to clearly mark digit I along with digit II (*Figure 1E* and *Figure 1—figure supplement 2*). Thus, although chondrification of the limb skeleton proceeds anterior to posterior in the zeugopod, digit specification as marked by *Sox9* expression in the autopod exhibits a pre-pattern that first emerges along the central axis marked by digit II.

### *Shh*-expressing cells in the presumptive ZPA are proximally restricted from the autopod and contribute exclusively to digit IV

Having established the spatiotemporal pattern of mesenchymal specification during skeletal formation, we next sought to investigate the expression of key genes involved in limb bud outgrowth and anterior-posterior patterning. For this, we optimized wholemount in situ hybridization such that we could accurately identify mesenchymal and ectodermal expression (*Figure 2—figure supplement 1*). During limb initiation in other tetrapods, *Sonic hedgehog* (*Shh*) is induced in the posterior limb mesenchyme (*Riddle et al., 1993*). Using an antisense-probe (~1000 bp) that was a gift from the Tanaka lab (IMP, Austria), we determined that *Shh* was first expressed at stage 45 in a posteriorly restricted domain and persisted through stage 49 in a posterior-proximal position (*Figure 2A*). Compared to anuran and other amniote forelimb buds where *Shh* expression precedes expression of *Hoxd11*, the onset of *Shh* expression in axolotls appeared relatively late and after *Hoxd11* expression (*Matsubara et al., 2017*; *Riddle et al., 1993*) (*Figure 2—figure supplement 2*). While the posterior *Shh* expression pattern was consistent with previous reports using relatively short probes at several stages (*Bickelmann et al., 2018*; *Imokawa and Yoshizato, 1997*; *Torok et al., 1999*), we also observed *Shh* expression in an anteriorly restricted domain at stages 46, 47 and 48 (*Figure 2A*). To support our in situ data, we divided stage 46–49 limb buds into posterior and anterior halves and collected RNA (*Figure 2—figure supplement 3A–C*). Using qRT-PCR, we observed *Shh* expression from the anterior and posterior compartments (*Figure 2—figure supplement 3D*). We also isolated full-length *Indian hedgehog* (*Ihh*) and aligned it with *Shh* revealing that our RNA probe shared 42% similarity between sequences (*Supplementary file 1*). Furthermore, we generated an RNA probe to *Ihh* and found spatiotemporal expression domains distinct from *Shh* expression localized to areas of skeletal ossification (*Figure 2—figure supplement 2*).

We next examined the spatiotemporal expression of the hedgehog receptor *Patched 1* (*Ptch1*) and effector molecule *Gli1*, both of which are direct targets of Shh-signaling (*Figure 2A–B*). *Ptch1* expression was localized in two broad domains corresponding to the posterior and anterior domains of *Shh* expression (*Figure 2A*). While we observed *Gli1* expression in an anterior domain similar to *Ptch1* expression, the posterior expression of *Gli1* appeared to localize more closely with *Ihh* (*Figure 2B* and *Figure 2—figure supplement 2*). Lastly, we examined the expression of the repressor form of *Gli3* which serves to restrict *Shh* to the posterior of the limb bud in mice and chickens and found a broad expression pattern across the anterior-posterior axis that excluded the presumptive ZPA at stages 45, 46 and 47 (*Figure 2* and *Figure 2—figure supplement 3D*).

Although *Shh* expression remains posterior in the forelimb buds of chicks and mice, it tracks distally into the future autopod where it maintains a close association with the AER (*Figure 2C–D*). In contrast, *Shh* expression in axolotl forelimb buds did not appear in the autopod (*Figure 2A,C–D*). Given this proximally restricted position of the *Shh* domain we asked whether *Shh*-expressing cells (or those close by) contribute to the digits. To track ZPA cells we injected DiI into the approximate position of *Shh* expression around stage 45 and monitored fluorescence till stage 53 (*Figure 2E*). Although our injections labeled *Shh*-expressing cells and nearby cells as well, we only observed cells migrating along the posterior limb margin and contributing to digit IV (*Figure 2E*). This data mimics labeling experiments in chick limbs where ZPA cells only give rise to the most posterior digit in the hindlimb and posterior margin of the forelimb (*Towers et al., 2008*; *Towers et al., 2011*). Thus, despite some late spatial expression differences, our data suggests a conserved role for *Shh* during forelimb development.

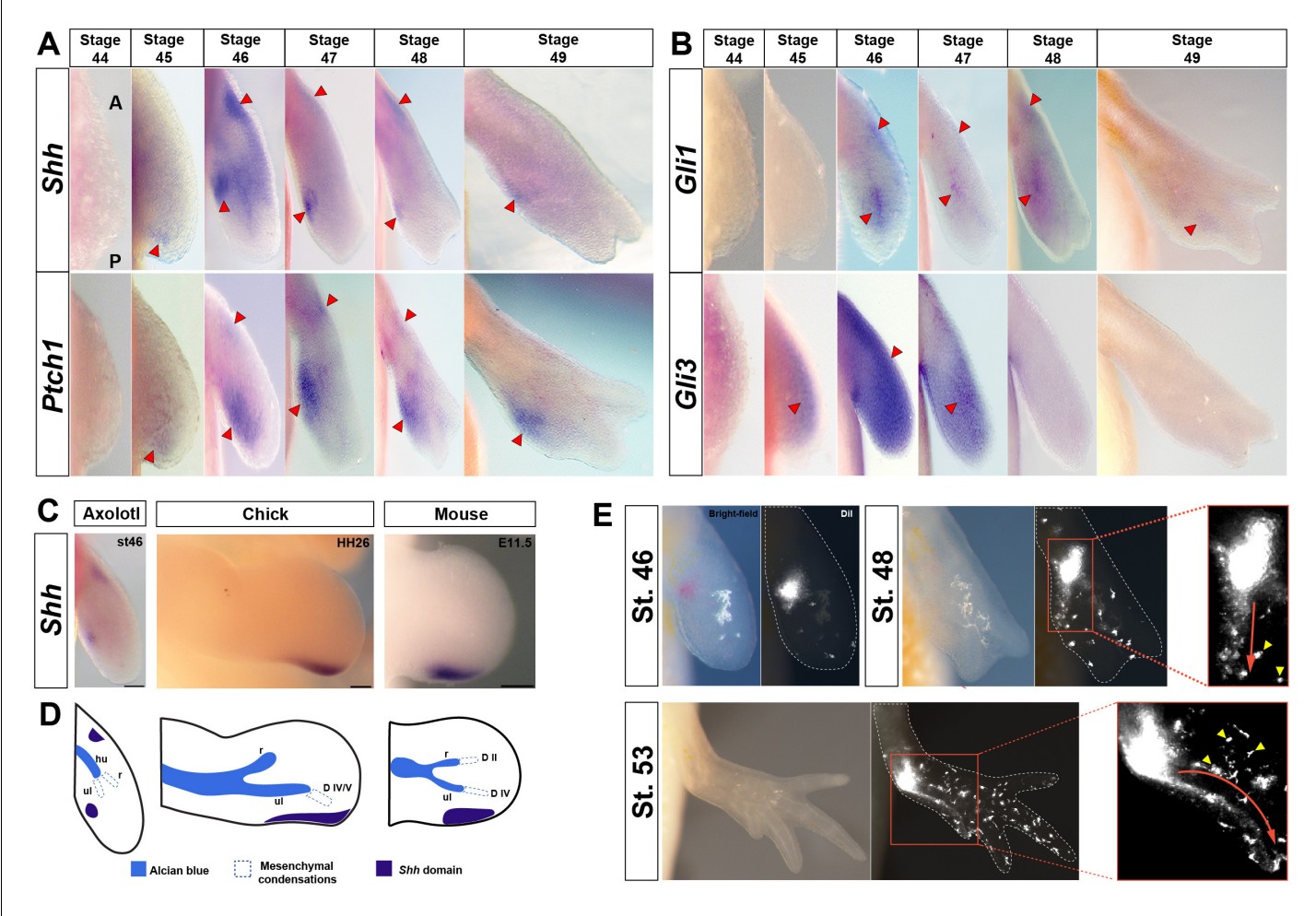

**Figure 2.** *Shh*-expressing cells in the presumptive ZPA are proximally restricted from the autopod and contribute progenitors exclusively to digit IV. (A–E) Dorsal views of stage 44–49 axolotl forelimbs with anterior (**A**) on top and posterior (**P**) on the bottom of each panel. Red arrows indicate expression. (**A**) *Shh* expression is first detected in a small posterior domain at stage 45 and persists through stage 49. Posterior expression is never detected in the autopod. An anterior domain of *Shh* expression is visible at stages 46–48. *Ptch1* expression overlaps with and surrounds *Shh* expression posteriorly and anteriorly. (**B**) *Gli1* expression is detected at stage 46 adjacent to, and slightly overlapping with, *Shh* and *Ptch1* expression. *Gli1* also exhibited a small anterior domain at stages 46–48. *Gli3* is expressed distally across the anterior-posterior axis between stages 45 and 48 with very weak expression overlapping the ZPA. (**C**) *Shh* domain in a stage 46 axolotl forelimb, HH26 chicken wing and E11.5 mouse forelimb. Scale bar = 100 µm (axolotl), 200 µm (chicken and mouse). (**D**) Schematic representation of *Shh* domain (indigo) compared to mesenchymal condensations (dotted) and Alcian blue stained cartilage condensations (blue) in a stage 46 axolotl forelimb, HH26 chicken wing (*Montero et al., 2017*) and E11.5 mouse forelimb (*Taher et al., 2011*). Cartilage condensations abbreviated: humerus (**hu**), radius (**r**) and ulna (**ul**). (**E**) DiI injections within the approximate *Shh* domain at stage 46 using light and fluorescence microscopy. Fluorescently labeled cells were followed through stage 53. Inset images at stages 48 and 53 show the migratory route of the DiI labeled mesenchymal cells (red arrow) to digit IV (n = 34). Yellow arrows represent pigment cells (autoflorescence) and ectodermal cells that have picked up the DiI.

The online version of this article includes the following figure supplement(s) for figure 2:

**Figure supplement 1.** Wholemount in situ hybridization (WISH) is optimized to detect gene expression in mesenchymal and ectodermal cells.
**Figure supplement 2.** *Hoxd11* is expressed prior to *Shh* expression at stage 44.
**Figure supplement 3.** Gene expression analysis of *Shh*, *Fgf8* and *Gli3* in anterior and posterior limb compartments.

## Amniote and anuran AER-specific *Fgf* ligands (*8, 9, 17*) are expressed exclusively in axolotl limb mesenchyme

Proximal-distal outgrowth of the limb bud and maintenance of Shh-signaling from the ZPA are regulated in tetrapods by the AER, and specifically by Fgf-signaling (*Lewandoski et al., 2000*; *Mariani et al., 2008*; *Niswander et al., 1993*; *Saunders, 1948*). Several *Fgfs* are expressed in the

anuran and amniote AER (i.e. *Fgf4, 8, 9, 17*), but *Fgf8* alone is required for cell survival and limb bud outgrowth (*Lewandoski et al., 2000*; *Sun et al., 2002*). Although an AER does not form during limb development in the direct developing frog *Eleutherodactylus coqui* (*Gross et al., 2011*) or the marsupial *Monodelphis domestica* (*Doroba and Sears, 2010*), AER-*Fgfs* are still restricted to, and expressed, in the ectoderm. Because salamanders lack a morphological AER, we asked if *Fgf4, 8, 9 and 17* were expressed in the axolotl limb bud ectoderm. In contrast to anurans and amniotes, we found that *Fgf8*, *Fgf9* and *Fgf17* were solely expressed in the mesenchyme (*Figure 3A–C* and *Supplementary file 1*). We could not consistently detect *Fgf4* during limb development and, when we did, it was expressed at very low levels in the mesenchyme only (*Figure 3—figure supplement 1*). At stage 44, we detected *Fgf8* in a broad mesenchymal zone directly beneath the ectoderm (*Figure 3A*). *Fgf8* expression persisted in the distal mesenchyme until stage 47 when it segregated into symmetrical domains within the dorsal and ventral mesenchyme (*Figure 3A–B*). Although *Fgf8* appeared to exhibit an anteriorly restricted expression pattern at stage 44, using qRT-PCR we found *Fgf8* expression in both the anterior and posterior compartment at stages 46, 47 and 49 (*Figure 2—figure supplement 3D*). *Fgf9* and *Fgf17* showed distally restricted expression at stages 45–46 and *Fgf9* appeared to have an additional proximal expression domain at stage 46 (*Figure 3C*). *Fgf17* appeared to have a posterior bias at stage 46 (*Figure 3C*). Consistent with what is known for amniotes and anurans, we found that *Fgf10* was broadly expressed in the distal mesenchyme at

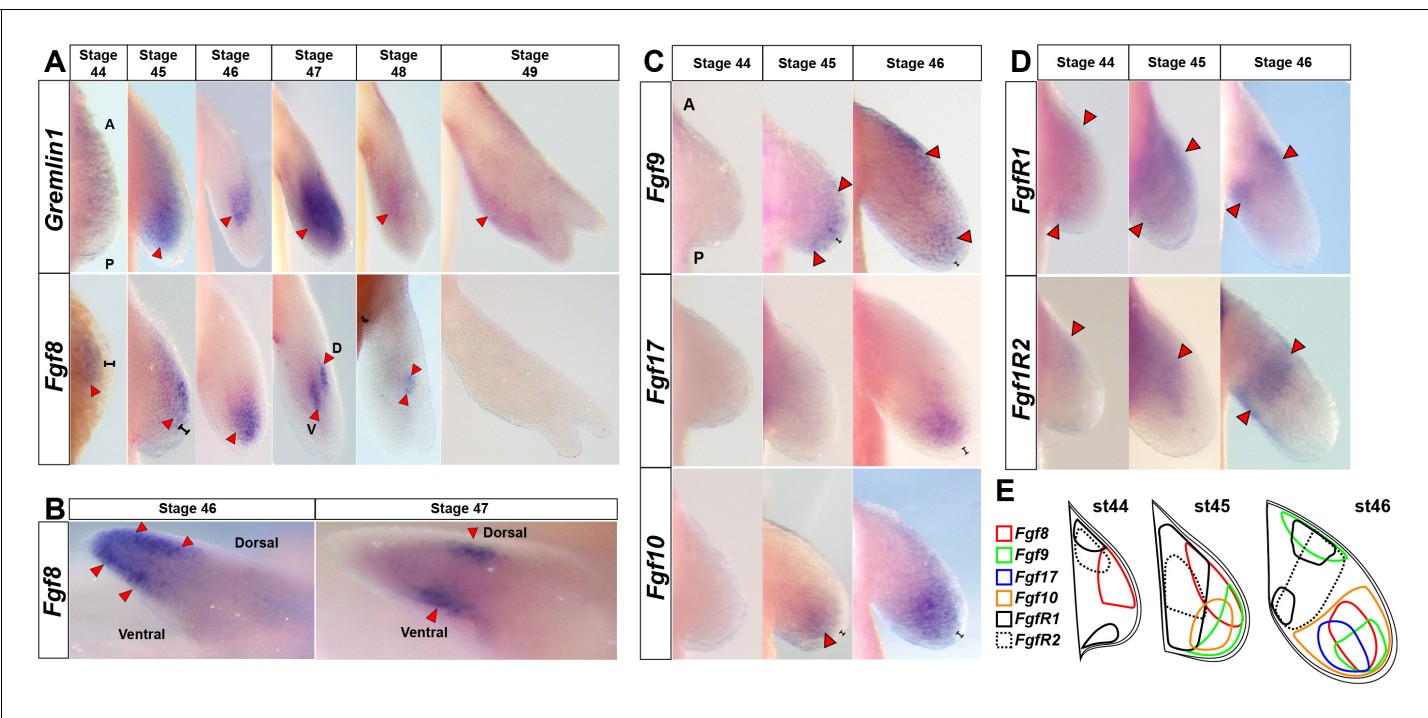

**Figure 3.** Amniote and anuran AER-specific *Fgf* ligands (*8, 9, 17*) are expressed exclusively in axolotl limb mesenchyme. (A, C–E) Dorsal views of stage 44–49 axolotl forelimbs with anterior (A) on top and posterior (P) on the bottom of each panel. Red arrows indicate expression domains. (A) *Gremlin1* and *Fgf8* expression at forelimb stages 44–49. *Gremlin1* is first expressed distally across the anteroposterior axis at stage 45. As the limb bud lengthens *Gremlin1* expression becomes centralized at the developing zeugopod and remains strongly expressed through stage 47. Between stages 48 and 49 *Gremlin1* expression becomes posteriorly restricted. *Fgf8* is expressed exclusively in the mesenchyme (stages 44–48). *Fgf8* expression is first detected at stage 44 with a slight anterior bias that expands distally until stage 46 and then shifts proximally. *Fgf8* expression was not detected at stage 49. (B) *Fgf8* expression at stage 46 begins to segregate dorsoventrally and ultimately separates into separate dorsal and ventral domains at stages 47–48. Anterior view of right limbs with dorsal side on top and ventral side on bottom. (C) *Fgf9* shows distal expression at stages 45–46 with an additional proximal domain at stage 46. *Fgf17* is expressed distally with a posterior bias at stage 46. *Fgf10* maintains distal mesenchymal expression at stages 45–46. (D) *FgfR1* and *FgfR2* are expressed proximally at stages 44–46. (E) Schematic representation of expression patterns for *Fgf8, Fgf9, Fgf17, Fgf10, FgfR1* and *FgfR2* at stages 44–46. Black brackets show the ectodermal layer.

The online version of this article includes the following figure supplement(s) for figure 3:

**Figure supplement 1.** *Fgf4* is expressed at extremely low levels in the limb mesenchyme.

stages 45–46 (*Figure 3C* and *Supplementary file 1*). We also examined expression of *Fgf receptors 1* and *2* (*FgfR1* and *FgfR2*). *FgfR1* was first expressed weakly at stage 44 and became more proximally restricted during stages 45–46 (*Figure 3D* and *Supplementary file 1*). At stage 46, the proximal-anterior domain of *FgfR1* overlaps with the *Fgf9* domain (*Figure 3C–D*). *FgfR2* showed weak proximal expression at stage 44 and was later expressed during stages 45–46 in a domain proximal to *Fgf8*, *9* and *17* (*Figure 3A–E* and *Supplementary file 1*). Lastly, we examined *Gremlin1* expression in developing salamander limbs, and observed strong mesenchymal expression until digits began condensing at stage 48 (*Figure 3A*). We also detected *Gremlin1* staining at stage 49 in the area destined to become digits III and IV (*Figure 3A*). Taken together, our expression analysis shows that key *Fgf* ligands normally expressed in the ectoderm during amniote and anuran limb development are instead, compartmentalized entirely in the limb mesenchyme (*Figure 3E*).

## Single-cell RNA-seq analysis supports spatial segregation of *Fgf* ligands and receptors

Our gene expression analysis suggested that *Fgf* ligands and their cognate receptors might be spatially segregated within the limb mesenchyme. To address this possibility, we analyzed single-cell RNA-seq (scRNA-seq) data from developing axolotl limbs that matched our limb stages 44 and 45 (see Materials and methods) (*Gerber et al., 2018*). Single-cell data acquired from these developing limbs included only mesenchymal cells (*Gerber et al., 2018*). Using principal component analysis, we first identified principal component 3 (PC3) as a model for the proximodistal (P-D) axis during stage 45 based on known proximal and distal marker genes (*Figure 4A* and *Figure 4—figure supplement 1*). Specifically, we found that proximal (*Meis1* and *Meis2*) and distal markers (*Fgf8* and *Hoxd11*) were expressed in cells on the opposite ends of contributors to PC3 (*Figure 4A–B* and *Figure 4—figure supplement 1*). Moreover, *Etv4* and *Dups6* which are direct targets of *Fgf8* were among the top 20 genes contributing to the distal end (*Hoxd11+*) of PC3 (*Figure 4A–B* and *Figure 4—figure supplement 1*). Consistent with expression patterns revealed by in situ hybridization, *Fgf* ligands such as *Fgf8*, *9*, *17* and *10* were all restricted to the distal end of PC3, while *Fgf* receptors *FgfR1-4* were restricted to the proximal end (*Figure 4A–B* and *Figure 4—figure supplement 1*). Despite seeing *FgfR1* expressed broadly across the modeled proximal-distal axis, our analysis showed that *FgfR1* was expressed to a greater extent in cells at the proximal end of PC3 (*Figure 4A–B*). Statistical analysis confirmed that *Fgf* ligands (n = 12) and *Fgf* receptors (n = 5) were separated along PC3 (Welch Two Sample t-test, $T = -4.1588$, p=0.0038). We also asked whether *Fgf* ligands and receptors might still be co-expressed in some cells and found that where *FgfR1* was expressed in the distal portion of PC3, some of these cells also expressed *Fgf8*, *Fgf9* and *Fgf17* (*Figure 4—figure supplement 2*). However, we found few to no cells co-expressing *FgfR2-4* and *Fgf8*, *9* and *17* (*Figure 4—figure supplement 2*). Considered together with our spatiotemporal expression analysis, our scRNA-seq analysis supports compartmentalization of Fgf-signaling within the developing limb mesenchyme and largely points to cellular segregation of ligand-receptor interactions.

## Early inhibition of Fgf-signaling reduces cell proliferation and leads to loss of posterior digits

In amniotes and anurans, a Shh-Grem-Fgf signaling loop regulates proximodistal outgrowth and maintains limb bud mesenchyme in a proliferative, undifferentiated, and multipotent state (*Globus and Vethamany-Globus, 1976*; *Reiter and Solursh, 1982*; *ten Berge et al., 2008*; *Towers et al., 2008*). To identify if a similar signaling loop exists in salamanders, we tested the functional requirement for Shh- and Fgf-signaling during axolotl limb development. First, in order to explore mesenchymal Fgf-signaling, we used the broad spectrum Fgf-receptor inhibitor (SU5402) that selectively binds to the intracellular kinase domain thereby inhibiting downstream signaling (*Mohammadi et al., 1997*). We treated axolotl embryos with SU5402 beginning prior to limb bud outgrowth from the flank (at stage 39) and then harvested limbs at stages 45, 46 or 54 (*Figure 5A*). We administered SU5402 based on a dose-response study and selected a maximum dose that could be delivered continuously which was not toxic to the developing animals (see Materials and methods). Limb buds harvested at stage 45 did not show a significant difference in limb size between treatment and control animals (one-tailed student's t-test, $T = -1.68637$, p=0.0514) (*Figure 5—figure supplement 1A*) whereas those harvested at stage 46 exhibited significantly smaller limbs (one-

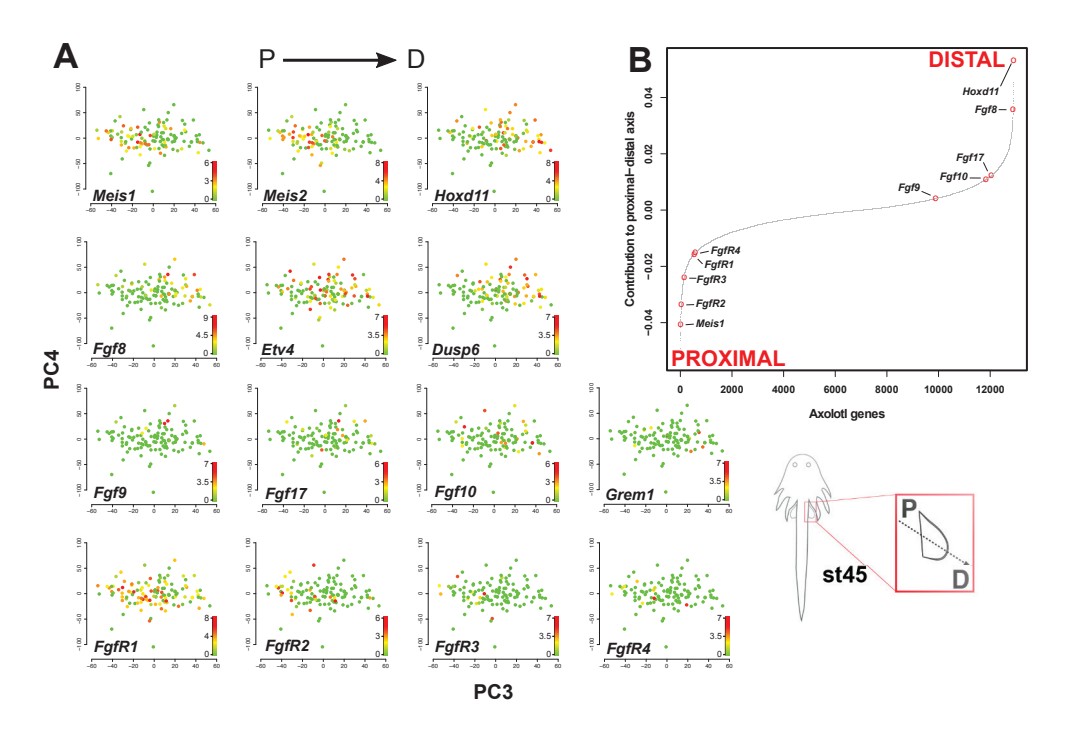

**Figure 4.** *Fgf* ligands and receptors are segregated along a modeled proximodistal axis. (**A**) Gene expression levels on PCA plot (PC3 vs PC4) for stage 45 limb bud mesenchyme. PC3 models the proximodistal axis of limb bud development exemplified by the expression of *Meis1* (proximal) and *Hoxd11* (distal). (**B**) Contributions of axolotl genes to PC3 highlighting the opposite directions of proximal (*Meis1*) and distal (*Hoxd11*) markers and that *Fgf* ligands and receptors are separated along the proximal-distal axis. The sigmoidal curve follows a normal distribution (Shapiro-Wilks normality test, $W = 0.96251$, $p<2.2e{-}16$). Color scale legends reflect log2(TPM) values.

The online version of this article includes the following figure supplement(s) for figure 4:

**Figure supplement 1.** Top twenty loadings of positive and negative axes of PC3 from PCA of stage 45 forelimbs.

**Figure supplement 2.** Cells expressing *FgfR1* also express *Fgf 8, 9, 10* and *17*.

tailed student's t-test, $T = -8.7759$, p<0.0001) (**Figure 5B**). Although we did find a small, but significant decrease in total animal length (snout to tail-tip length) following SU5402 treatment (one-tailed student's t-test, $T = -4.52$, p=0.0007) (**Figure 5—figure supplement 1B**), this did not account for the size differences between treatment and control limbs ($T = -8.1$, p<0.44). To test the efficacy of Fgf-signaling inhibition, we determined expression of the Fgf-signaling targets *Etv1* and *Etv4* (**Kawakami et al., 2003**) (**Figure 5C–E**). In response to Fgf inhibition, we were unable to detect *Etv1* expression in the limb bud at stages 45 or 46 and *Etv4* expression was barely detectable (**Figure 5C–D**). qRT-PCR confirmed that both targets were significantly down-regulated at stage 46 (**Figure 5E**).

To determine if Fgf-signaling controls Shh-signaling, we quantified *Shh* expression in response to *Fgf* inhibition and found that it was reduced compared to control limbs (**Figure 5E**). Given the relatively small number of *Shh*-expressing cells at these early time points, we asked if reduced *Shh* transcription translated into reduced pathway activity as assessed by *Ptch1* expression (**Figure 5E–G**). Using qRT-PCR and in situ, we found that *Ptch1* was not reduced in treated limb buds suggesting that Shh pathway activity remained normal (**Figure 5E–G**). Interestingly, while *Gremlin1* is inhibited by AER-Fgf-signaling in amniotes (**Merino et al., 1999**; **Verheyden and Sun, 2008**), we found that *Gremlin1* expression was virtually eliminated in SU5042-treated limbs (**Figure 5E–G**).

Loss of AER-Fgf-signaling during mouse limb development does not affect cell proliferation, but produces smaller limbs and proximal truncation due to increased proximal cell death (**Mariani et al., 2008**; **Sun et al., 2002**). We used Lysotracker to mark dying cells (**Mariani et al., 2008**; **Seifert et al., 2009**) and during normal limb development we did not detect any dying cells

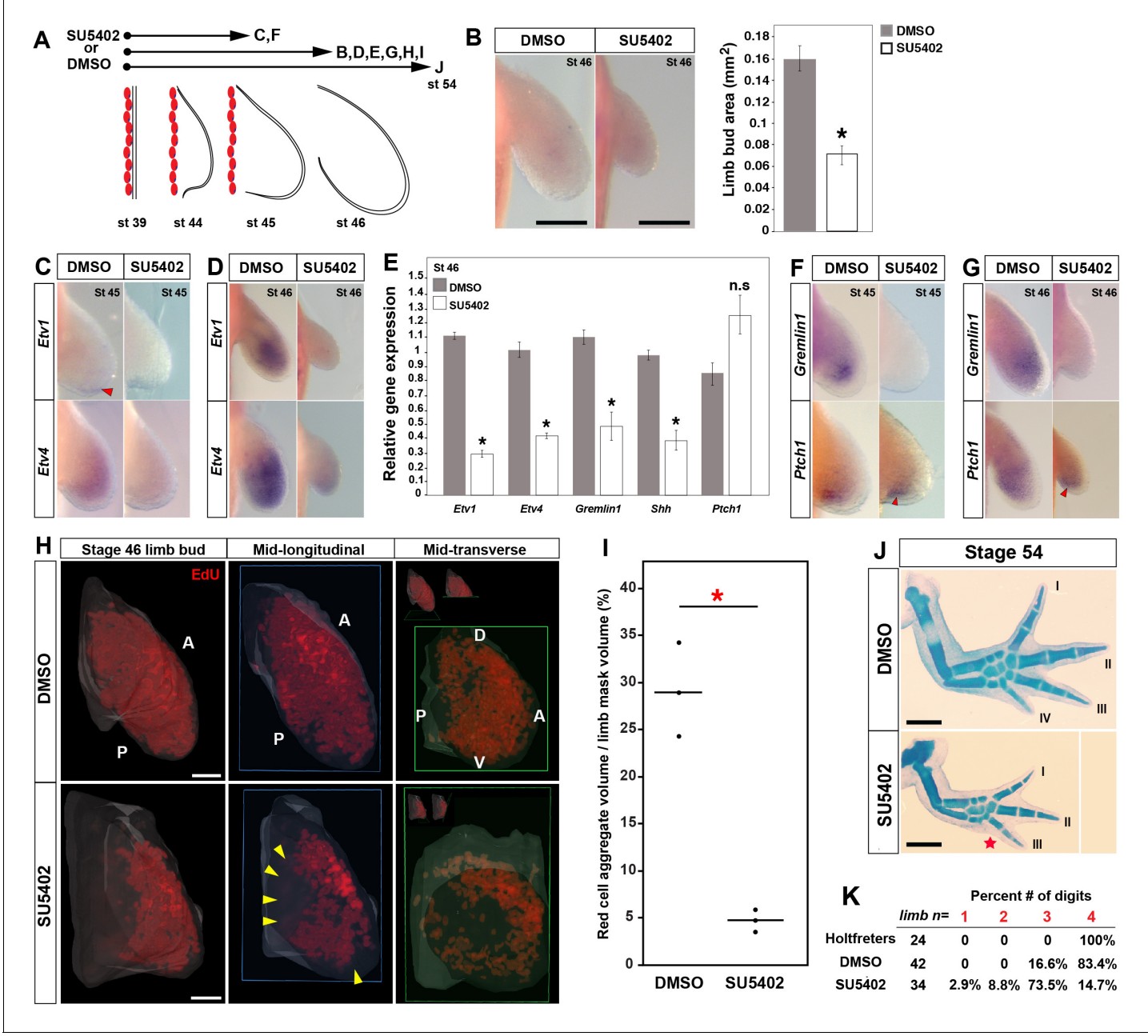

**Figure 5.** Early inhibition of Fgf-signaling reduces cell proliferation and leads to loss of posterior digits. (A) Design for SU5402 and DMSO treatments. Capital letters refer to harvest stage and figure panel. Red ovals depict dorsal muscle blocks. (B) Limb bud size (area) at stage 46 shows a significant decrease in SU5402-treated larvae (one-tailed student's t-test, $T = -8.7759$, asterisk represents p<0.0001, n = 5 for DMSO and n = 6 for SU5402). Error bars represent standard error of mean. Scale bar = 100 μm. (C–D) SU5402 treatment efficiently down-regulates Fgf-signaling targets *Etv1* and *Etv4* at stages 45 and 46. (E) qRT-PCR analysis of stage 46 limb buds from DMSO (control) and SU5402 treatments indicates down-regulation of *Etv1*, *Etv4*, *Gremlin1* and *Shh*. In contrast, *Ptch1* expression appeared unaffected in the treated limbs (two tailed t-test; *Etv1: T = -19.13*, p<0.0001; *Etv4: T = -9.79*, p=0.0006; *Gremlin1: T = -4.14*, p=0.014; *Shh: T = -6.95*, p=0.0022 and *Ptch1: T = 2.44*, p=0.070; n = 3 for DMSO and n = 3 for SU5402; asterisk depicts significant p-values; n.s = not significant). Relative gene expression depicted as $2^{-\Delta\Delta Ct}$ values with *GAPDH* as the housekeeping gene. Error bars represent standard error of mean. (F–G) SU5402 treatments cause a down-regulation of *Gremlin1* while *Ptch1* expression remained unaffected at both stages. (H–I) SU5402-treated stage 46 limbs show a significant decrease in the EdU-positive cells. Lightsheet images are depicted as volume rendered red cell aggregates within a hand drawn limb mask (dorsal view with anterior **A** on top and posterior **P** on the bottom), mid-longitudinal sections (blue box = plane of section) of the volume rendered limbs and mid-transverse sections (green box = plane of section) of the volume rendered limbs (H). Abbreviations: dorsal (**D**) and ventral (**V**) in the mid-transverse sections. The in-set images on top of the mid-transverse sections depict the orientation of the limb and the plane of section. Cell proliferation is down-regulated throughout the limb, and very few proliferating cells were present in the proximal and distal parts of treated limbs (yellow arrows). Scale bar = 100 μm. (I) Statistical comparisons within control (DMSO)

*Figure 5 continued on next page*

*Figure 5 continued*

and treatment (SU5402) groups were determined by one-way ANOVA (Kruskal-Wallis tests). n = 3 for DMSO and SU5402. Horizontal bars represent median values; p<0.05. Asterisk depicts significant p-value. (J) Alcian blue stained DMSO (control) and SU5402-treated stage 54 limbs. (K) Drug-treated animals had smaller limbs with 73.5% lacking posterior digit IV (red star in J indicates position of missing digit). Scale bar = 1 mm.

The online version of this article includes the following figure supplement(s) for figure 5:

**Figure supplement 1.** Effect of DMSO, SU5402, ethanol and cyclopamine treatments on limb size at stage 45 and snout to tail-tip length at stage 46.

**Figure supplement 2.** Cell death is not evident during normal axolotl forelimb development or in response to inhibition of Fgf- or Shh-signaling.

(*Figure 5—figure supplement 2*). Similarly, when we inhibited Fgf-signaling we did not detect dying cells in limb buds (*Figure 5—figure supplement 2*). We next assessed if smaller limbs might have resulted from alterations in cell proliferation (*Figure 5H–I*). Inhibiting Fgf-signaling as above, we examined actively proliferating cells (EdU+) in treated and control stage 46 limb buds using light-sheet microscopy (*Figure 5H–I*). We calculated the proliferative population as a fraction of total limb volume and observed an 83% decrease in actively proliferating cells (one-way ANOVA, Kruskal-Wallis test, p<0.05) (*Figure 5I*). While the decrease in proliferating cells appeared proportionally across the anteroposterior axis, we noted that proliferating cells were nearly absent from the proximal limb bud and from a small distal domain in treated limbs (*Figure 5H*).

To determine the ultimate effect of Fgf inhibition on limb development, we assessed the effect of treating embryos from before limb outgrowth until all four digits appeared in control limbs at stage 54 (*Figure 5J–K*). Analyzing the forelimb skeleton, we found that inhibiting Fgf-signaling led to smaller, but nearly complete limbs which generally lacked posterior digit IV (*Figure 5J–K*). Specifically, 73.5% of SU5402-treated embryos exhibited this phenotype (25/34) while ~11% (4/34) had fewer than three digits (*Figure 5I*). In those cases where more than one digit was lost, the missing digits were the next most posterior in sequence. These results reveal that Fgf-signaling regulates mesenchymal proliferation, but not cell survival during axolotl limb development. Thus, inhibiting Fgf-signaling leads to smaller limbs and loss of posterior digits, a result consistent with colchicine treatment of developing axolotl and *Xenopus* larva (*Alberch and Gale, 1983*). Lastly, Fgf-signaling regulates *Gremlin1* expression, whereas Shh-signaling is relatively independent of Fgf-signaling.

## *Sonic hedgehog* signaling regulates *Fgf8* expression during axolotl limb development

While our results above suggest that a Shh-Grem-Fgf signaling loop does occur during axolotl limb development, we sought to examine if Fgf-signaling was reliant on Shh-signaling. Using cyclopamine to inhibit Shh signal transduction, previous work showed that Shh-signaling controls anterior-posterior patterning of the zeugopod and autopod during axolotl limb development, a role consistent with ZPA function in other tetrapods (*Stopper and Wagner, 2007*). Cyclopamine-treated axolotl limbs phenocopied *Shh*$^{-/-}$ mouse limbs with significant proximodistal outgrowth, fusion of the radius/ulna, and almost complete loss of the autopod (*Chiang et al., 2001*). To analyze the interaction of Shh- and Fgf-signaling during limb bud outgrowth, we treated stage 39 larvae with cyclopamine for 10 days and analyzed the limb buds at stages 45 and 46 (*Figure 6A*). Limb buds harvested at stage 45 did not show a significant difference in limb size between treatment and control animals (one-tailed student's t-test, $T = 1.43$, p=0.909) (*Figure 5—figure supplement 1C*), whereas at stage 46, limb buds from cyclopamine-treated larvae were significantly smaller compared to control treated limbs (one-tailed student's t-test, $T = 8.36$, p<0.0001) and this effect was independent of ($T = 0.03$, p=0.975) a small, but significant decrease in animal length (one-tailed student's t-test, $T = 3.87$, p<0.0016) (*Figure 6B* and *Figure 5—figure supplement 1D*). Analyzing *Ptch1* expression by qRT-PCR and in situ hybridization, we found that cyclopamine treatment efficiently inhibited hedgehog signaling in limb buds (*Figure 6C–E*). Using qRT-PCR we saw a significant reduction in *Fgf8* and *Gremlin1* expression at stage 46 and using in situ we were unable to detect these genes at either stage 45 or 46 (*Figure 6B–D*). These data place *Fgf8* and *Gremlin1* downstream of Shh-signaling during axolotl limb development. We also tested whether Shh-signaling controlled cell survival and cell proliferation. Similar to our results using SU5042, when we inhibited Shh-signaling we did not observe cell death in developing limb buds (*Figure 5—figure supplement 2*). However, we did find that loss of Shh-signaling led to a 53% reduction in cell proliferation (*Figure 6F–G*). Whereas Fgf-

inhibition led to proportionally smaller limbs, Shh-inhibition led to a dramatic loss of cell proliferation in the distal limb bud (one-way ANOVA, Kruskal-Wallis test, p<0.05) (*Figure 6F*). Together, our findings place *Fgf8* downstream of Shh-signaling and suggest that Shh-signaling also controls cell proliferation, although more specifically in the distal limb bud where digit progenitors reside.

## Discussion

### Comparing forelimb development between urodeles, anurans and amniotes

Urodeles were among the first vertebrates used to study limb field specification and morphogenesis (*Harrison, 1918*; *Stocum and Fallon, 1982*; *Swett, 1937*). Although chick and mouse embryos largely replaced urodeles as model systems to study limb development, a perpetual fascination with understanding the molecular basis of limb regeneration has resurrected interest into amphibian limb development (*Gerber et al., 2018*; *Keenan and Beck, 2016*; *Stocum, 1975*). Elucidating the cellular and molecular basis of limb development in urodeles and other amphibians also has important implications for our understanding of how limbs evolved (*Alberch and Gale, 1983*; *Fröbisch and Shubin, 2011*; *Stopper and Wagner, 2005*). Despite deep homology among tetrapod limbs, biologists have long recognized several unique aspects of urodele limb development (*Holmgren, 1933*). For example, although the limb field in urodeles is established in the gastrula, the limb bud does not emerge from the flank until much later in a free-swimming larva. Unlike amniotes, this situation demonstrates a level of autonomous development that is temporally de-coupled from limb specification and patterning of the main body axis (*Stocum and Fallon, 1982*). Retinoic acid (RA) generated from the somites in chicks and mice diffuses into the lateral plate mesoderm where it permits correct spatiotemporal induction of *Tbx5* (*Cunningham et al., 2013*; *Nishimoto et al., 2015*; *Stephens and McNulty, 1981*). In contrast, during anuran limb bud development which occurs in larval animals (similar to urodeles), retinoic acid (RA) appears to be generated autonomously in the forelimb bud with *raldh2* expressed proximally and *cyp26b* distally (*McEwan et al., 2011*). In addition to limb heterochrony, urodele limb buds do not form an apical ectodermal ridge (AER) (*Sturdee and Connock, 1975*; *Tank et al., 1977*) and exhibit preaxial dominance of the limb skeleton (*Shubin and Alberch, 1986*). Lastly, urodeles can regenerate an entire limb, something no other group of tetrapods can do as adults (*Table 1*). Thus, while these aspects of urodele limb development challenge our notion of an inclusive vertebrate limb development model (*Zeller et al., 2009*) they also beg the question; does the molecular machinery directing limb morphogenesis exhibit critical differences when compared to amniotes and anurans?

In this study, we examined several key aspects of salamander limb development as they relate to patterning and outgrowth of the tetrapod limb. In doing so, we considered salient features of salamander forelimb development as they compare to *Xenopus*, chickens and mice (summarized in *Table 1*). First, we confirm previous reports that axolotls lack a morphological AER. Second, using *Sox9* expression, we show that digit specification occurs first along the metapterygial axis of the limb with digit II, and then proceeds postaxial with digits I, III and IV. However, using Alcian blue staining as a proxy for cartilage condensation, we show that digits I and II differentiate simultaneously and are followed in sequence by digits III and IV. Thirdly, we show that *Shh* is restricted posteriorly, and although *Shh* expression does not overlap with the autopod, ZPA cells contribute to digit IV. Together with cyclopamine experiments these data support Shh-signaling as a key mediator of anterior-posterior patterning. Fourthly, we focused on the cellular source of Fgf-signaling and find that, in contrast to anurans and amniotes where reciprocal signaling is compartmentalized between the limb ectoderm and mesenchyme, *Fgf* ligands (*Fgf8, 9, 17, 10*) and receptors (*FgfR1-4*) are all expressed solely in the mesenchyme. By functionally testing the requirement for Fgf-signaling using a broad Fgf-receptor antagonist (SU5042), we demonstrate that Fgf-signaling regulates limb size by controlling cell proliferation across all three limb axes. Again, this stands in contrast to anurans and amniotes where Fgf-signaling regulates cell survival and cellular differentiation along the proximal-distal axis. Another key finding from these experiments is that Fgf-signaling regulates *Gremlin1*, whereas Shh-signaling is maintained in the face of Fgf-inhibition. While these results strongly suggest that a Shh-Grem-Fgf signaling loop is not present during salamander limb development, they do show that *Fgf8* expression is dependent on Shh-signaling. Together, our results show Shh-signaling

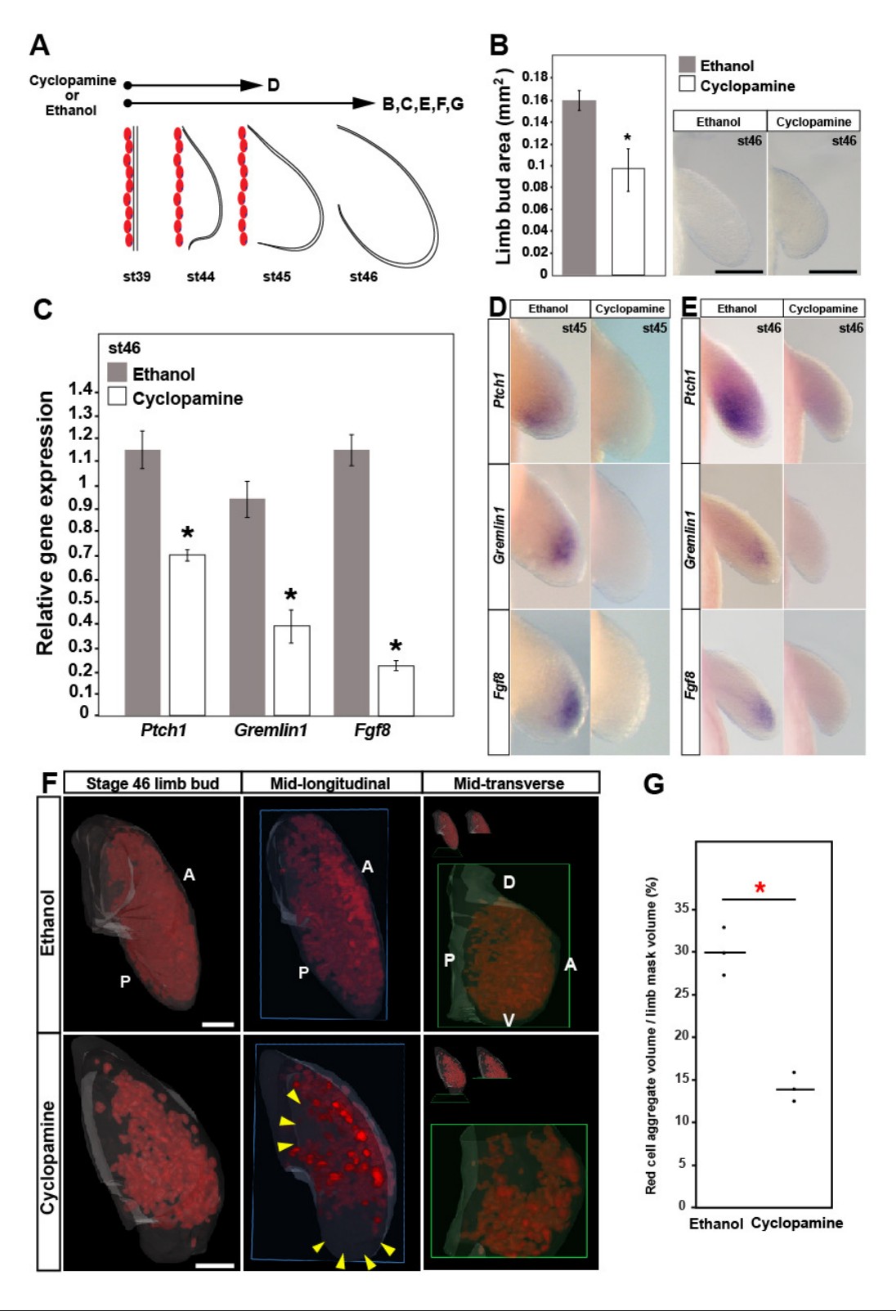

**Figure 6.** *Sonic hedgehog* signaling regulates *Fgf8* expression and distal cell proliferation during axolotl limb development. (A) Design for cyclopamine and ethanol treatments. Capital letters refer to harvest stage and figure panel. Red ovals depict dorsal muscle blocks. (B) Limb bud size (area) at stage 46 shows a significant decrease in cyclopamine-treated larvae (one-tailed student's t-test, *T* = 8.36, p<0.0001, n = 6 each for ethanol and cyclopamine). Scale bar = 100 µm. (C) Real-time PCR analysis of *Ptch1*, *Gremlin1* and *Fgf8* in stage 46 limb buds from ethanol (control) and cyclopamine treatments

*Figure 6 continued on next page*

*Figure 6 continued*

(one tailed t-test; asterisk depicts significant p-values; *Ptch1: T* = 4.95, p=0.0039; *Gremlin1: T* = 5.97, p=0.002 and *Fgf8: T* = 11.22, p=0.0002; n = 3 for ethanol and cyclopamine). Relative gene expression depicted as $2^{-\Delta\Delta Ct}$ values with *RPL32* as the housekeeping gene. Error bars represent standard error of mean. (D–E) Cyclopamine treatment efficiently down-regulates *Ptch1* and therefore Shh-signaling. Cyclopamine treatment also down-regulates *Gremlin1* and *Fgf8*. (F–G) Cyclopamine-treated stage 46 limbs show a significant decrease in EdU-positive cells. Lightsheet images depicted as volume rendered red cell aggregates within a hand drawn limb mask (dorsal view with anterior **A** on top and posterior **P** on the bottom), mid-longitudinal sections (blue box = plane of section) of the volume rendered limbs and mid-transverse sections (green box = plane of section) of the volume rendered limbs. Abbreviations: dorsal (**D**) and ventral (**V**) in the mid-transverse sections. The in-set images on top of the mid-transverse sections depict the orientation of the limb and the plane of section. There is a domain specific loss of cell proliferation in the proximal and distal parts of the treated limbs (yellow arrows). Scale bar = 100 µm. (G) Comparisons within control (ethanol) and treatment (cyclopamine) groups were determined by one-way ANOVA (Kruskal-Wallis tests); n = 3 for ethanol and cyclopamine. Horizontal bars represent median values; p<0.05. Asterisk depicts significant p-value.

from the ZPA has maintained its core function while de-coupling itself from Fgf-signaling and that Fgf-signaling has evolved to regulate cell proliferation in the limb.

## Skeletal differentiation during axolotl forelimb development

Our analysis of skeletogenesis shows that axolotl digits appear to be specified in a different order than they differentiate. Taking advantage of subtle variations in within animal and between animal staging, our analysis using *Sox9* expression shows a 2>1>3>4 pattern of digit specification. However, when we observed digit differentiation using Alcian blue to label condensing cartilage we always found digit I and II appearing together followed in sequence by digits III and IV. This pattern of digit differentiation is consistent with that observed by other investigators using histological preparations (*Fröbisch, 2008*; *Shubin and Alberch, 1986*) or Alcian blue (*Nye et al., 2003*). These findings support independent molecular control of digit specification and differentiation and hint at the wide diversity in ontogenetic and heterochronic shifts that have occurred in the limb during urodele evolution (*Blanco and Alberch, 1992*; *Franssen et al., 2005*). Similarly, *Sox9* expression shows spatiotemporal variability across urodeles (*Kerney et al., 2018*) and this underscores the need for more comparative limb studies to better understand the ancestral condition. While preaxial vs. postaxial dominance of skeletal formation clearly separates urodele limb development from anurans and amniotes, it is likely this difference points to alterations in the upstream genetic control of digit specification involving *Fgfs*, *Bmps*, and retinoic acid (*Montero et al., 2017*; *ten Berge et al., 2008*).

## A conserved role for *Sonic hedgehog* signaling during vertebrate limb development

With respect to anterior-posterior patterning, our data show that the members of the hedgehog signaling pathway are expressed in a mesenchymal pattern consistent with other tetrapods studied to date. However, our results also reveal three important differences. First, our data reveals that *Shh* expression appears relatively late during limb bud outgrowth and after *Fgf8* and *Hoxd11* are expressed. Second, salamander forelimb buds exhibit a proximal-anterior domain of *Shh* expression that emerges as the axolotl limb bud begins to bend posteriorly; a domain which is not found during normal limb development in amniotes and anurans. Third, posterior *Shh* expression corresponding to the ZPA remains in a relatively small proximal domain (*Torok et al., 1999*) rather than the elongated expression domain found in other tetrapods that extends the proximal-distal length of the autopod (*Matsubara et al., 2017*; *Riddle et al., 1993*; *Shapiro et al., 2003*). With respect to the first point, previous work has demonstrated that flank tissue surrounding the limb field plays an important role in specifying the anteroposterior axis and thus, the temporal appearance of *Shh* expression may be less important than its posterior induction (*Stocum and Fallon, 1982*). In chick limbs, *Hoxd11* can be induced by *Shh,* but only in proximity to the AER or in the presence of AER-Fgfs (*Laufer et al., 1994*) and in *Shh* KO mice *Hoxd11* is still expressed in the limb bud (*Chiang et al., 2001*). While intriguing, the transient appearance of an anterior *Shh* domain is difficult to explain. We did not detect an anterior necrotic zone (*Figure 5—figure supplement 2*), and *Shh* inhibition did not lead to increased cell death in this region. Moreover, *Shh* inhibition produced phenotypes consistent with other amniote models of limb development (*Chiang et al., 2001*; *Scherz et al., 2007*; *Stopper and Wagner, 2007*; *Towers et al., 2008*; *Vargas and Wagner, 2009*; *Zhu et al., 2008*). When Shh-signaling is inhibited in axolotls with cyclopamine, zeugopodial and

**Table 1.** Salient features of forelimb development between axolotl, *Xenopus*, chicken and mouse.

| Feature | Axolotl | *Xenopus* | Chicken | Mouse | References |
|---|---|---|---|---|---|
| Autopod skeletal differentiation | Preaxial dominance[1] | Postaxial dominance[1] | Postaxial dominance[1] | Postaxial dominance[1] | [1] *Shubin and Alberch, 1986* |
| Location of ZPA domain | Posterior. Excluded from autopod[1,*] | Posterior. Extends into autopod[2] | Posterior. Extends into autopod[3] | Posterior. Extends into autopod[4] | [1] *Torok et al., 1999* [2] *Endo et al., 1997* [3] *Riddle et al., 1993* [4] *Echelard et al., 1993* * This study |
| Contribution of ZPA cells to posterior digit(s) | Yes (DiI labeling)[*] | ? | Yes (DiI labeling)[1] No (GFP grafting)[2] | Yes (Genetic labeling)[3] | [1] *Towers et al., 2008* [2] *Towers et al., 2011* [3] *Harfe et al., 2004* * This study |
| Shh-signaling during limb development | Key mediator of anterior-posterior patterning[1] | Key mediator of proximal-distal and anterior-posterior patterning[2] | Key mediator of anterior-posterior patterning[3] | Key mediator of anterior-posterior patterning[4, 5] | [1] *Stopper and Wagner, 2007* [2] *Stopper et al., 2016* [3] *Ros et al., 2003* [4] *Chiang et al., 1996* [5] *Chiang et al., 2001* |
| Morphological AER | No[1] | Transient[2] | Yes[3] | Yes[4] | [1] *Tank et al., 1977* [2] *Tarin and Sturdee, 1971* [3] *Saunders, 1948* [4] *Wanek et al., 1989* |
| Molecular AER | No[*] | Yes | Yes | Yes | * This study |
| Location of AER-specific *Fgfs* (4,8,9,17) | Mesenchyme[*] | AER[1] | AER[2, 3, 4] | AER[2, 5, 6, 7, 8, 9] | [1] *Christen and Slack, 1997* [2] *Mahmood et al., 1995* [3] *Duprez et al., 1996* [4] *Havens et al., 2006* [5] *Ohuchi et al., 1994* [6] *Crossley and Martin, 1995* [7] *Niswander and Martin, 1992* [8] *Sun et al., 2002* [9] *Sun et al., 2000* * This study |
| Location of *Fgf* receptors | *FgfR1-4* expressed exclusively in the mesenchyme[*] | ? | *FgfR1IIIc, FgfR2IIIb, FgfR3IIIb* expressed in the ectoderm and *FgfR1IIIb, FgfR2IIIc, FgfR3IIIb/c* and *FgfR4* expressed in the mesenchyme[1, 2] | *FgfR1IIIb* and *FgfR2IIIb* expressed in the ectoderm and *FgfR1IIIc, FgfR2IIIc, FgfR3c* and *FgfR4c* expressed in the mesenchyme[3, 4, 5] | [1] *Havens et al., 2006* [2] *Sheeba et al., 2010* [3] *Min et al., 1998* [4] *MacArthur et al., 1995* [5] *Ornitz and Itoh, 2015* * This study |
| Fgf-signaling during limb development | Controls cell proliferation and limb size[*] | ? | Regulates proximodistal patterning, cell survival and cellular differentiation[1, 2, 3,4] | Regulates proximodistal patterning, cell survival and cellular differentiation[5, 6] | [1] *Saunders, 1948* [2] *Summerbell, 1974* [3] *Janners and Searls, 1971* [4] *Dennis Summerbell, 1977* [5] *Sun et al., 2002* [6] *Mariani et al., 2008* * This study |
| Positive regulation of *Gremlin1* by Fgf-signaling in the limb | Yes[*] | ? | No[1] | No[2] | [1] *Merino et al., 1999* [2] *Verheyden and Sun, 2008* * This study |
| Positive regulation of Shh-signaling by Fgf-signaling in the limb | No[*] | ? | Yes[1] | Yes[2] | [1] *Crossley et al., 1996* [2] *Lewandoski et al., 2000* * This study |

*Table 1 continued on next page*

*Table 1 continued*

| Feature | Axolotl | *Xenopus* | Chicken | Mouse | References |
|---|---|---|---|---|---|
| Positive regulation of Fgf-signaling by Shh-signaling in the limb | Yes[*] | ? | Yes[1] | Yes[2] | [1] *Laufer et al., 1994*<br>[2] *Harfe et al., 2004*<br>[*] This study |
| Limb regeneration | Regenerates complete limb through adulthood[1, 2, 3] | Regenerates complete limb through larval stage 53[4] | Does not regenerate[5] | Does not regenerate. [*]Regenerates digit tip[6] | [1] *Young et al., 1983a*<br>[2] *Young et al., 1983b*<br>[3] *Monaghan et al., 2014*<br>[4] *Dent, 1962*<br>[5] *Muneoka and Sassoon, 1992*<br>[6] *Borgens, 1982* |

autopodial skeletal elements are lost in a posterior to anterior direction that is dependent on when cyclopamine is administered (*Stopper and Wagner, 2007*). A similar phenotype occurs in chick embryos where elegant work demonstrated that Shh-signaling controls digit progenitor specification and limb bud growth (*Towers et al., 2008*). When coupled with the results of *Shh* inhibition (*Stopper and Wagner, 2007*), our observations that Shh-signaling controls distal cell proliferation and that *Shh*-expressing cells contribute to digit IV, our findings support a conserved role for Shh-signaling as it pertains to specifying digit progenitors during limb development across all tetrapods. Future experiments ectopically expressing Shh using beads or virus will serve as a means to test downstream targets of Shh-signaling, as will transgenic elimination of *Shh* during development.

## Developing urodele limbs lack a morphological and molecular AER

In amniotes and anurans, limb bud outgrowth and proximal-distal patterning are controlled by the AER via expression of specific *Fgf* ligands (e.g. *Fgf4, 8, 9, 17*) (*Cohn et al., 1995*; *Lewandoski et al., 2000*; *Mariani et al., 2008*; *Ohuchi et al., 1997*; *Sun et al., 2002*). The AER maintains ZPA activity (*Vogel and Tickle, 1993*) and AER-*Fgfs* and *Fgf*-regulated *Etvs* are essential in inducing and maintaining posterior expression of *Shh* in the ZPA (*Niswander, 2002*; *Zhang et al., 2009*). Furthermore, *Gremlin1* acts as an intermediary between these signaling centers where it relays signals from the ZPA to control Bmp-signaling and maintain the AER (*Zeller et al., 2009*; *Zuniga, 2015*). In stark contrast to anurans and amniotes, our data represents the first instance of a tetrapod lacking *Fgf* ligand and receptor expression in the developing limb bud ectoderm. Although previous studies suggested that *Fgf8* was expressed early in salamander limb bud ectoderm (*Han et al., 2001*), a result consistent with studies in *Monodelphis domestica* (*Doroba and Sears, 2010*) and *Eleutherodactylus coqui* (*Gross et al., 2011*) which lack an AER but exhibit ectodermal *Fgf8* expression, our findings during development show this is not the case. Moreover, our expression patterns for *Fgf8* are consistent with recent data from regenerating axolotl limbs where *Fgf8* expression was found restricted to the mesenchyme (*Nacu et al., 2016*). Consistent with our spatiotemporal expression results, analysis of scRNA-seq data from stage 45 axolotl limb buds (*Gerber et al., 2018*) showed similar patterns along a modeled proximodistal axis. Co-expression analysis of *Fgf* ligands and receptors at stage 45 provide evidence that some cells expressing *FgfR1* also express several *Fgf* ligands. Given the relatively small number of cells analyzed in this dataset (<200), a more complete scRNA-seq analysis across multiple time points will help address the complexity of Fgf-signaling and whether mesenchymal cells secreting *Fgf* ligands respond in an autocrine fashion. Our experiments also call into question what role, if any, the ectoderm plays during urodele limb development. For instance, experiments in *Pleurodeles waltl* showed that early stage forelimb bud mesoderm could autonomously develop a fully formed limb when grafted under heterologous flank epidermis (*Lauthier, 1985*). A clear functional role of the ectoderm awaits genetic manipulation since manual removal of the ectoderm results in rapid regeneration.

## Fgf-signaling regulates cell proliferation and limb size

Our experiments using the *Fgf* receptor antagonist SU5402 sought to test if spatial re-location of *Fgf* ligands and receptors affected the established function of Fgf-signaling during anuran and amniote limb bud development. SU5402 inhibition (beginning before limb bud outgrowth) revealed that the primary function of Fgf-signaling in axolotl limb development is to regulate cell proliferation throughout the limb bud. It does not, however, appear to control cellular differentiation, cell survival or proximodistal patterning. Removal of the AER in chickens (*Saunders, 1948*) leads to a stage dependent loss of skeletal elements in a proximal to distal direction and combined genetic deletion of *Fgf8/Fgf4/Fgf9* in the mouse AER (*Mariani et al., 2008*) leads to a complete loss of the stylopod, zeugopod and autopod. In contrast, broadly inhibiting Fgf-signaling from the outset of axolotl limb development phenocopies urodele limbs treated with the mitotic inhibitor colchicine where in both cases, the most posterior digit fails to form in an otherwise smaller, but normal limb (*Alberch and Gale, 1983*). These results echo experiments in chickens where pharmacological inhibition of cell proliferation using trichostatin A, colchicine, or vinblastine leads to loss of anterior digits, which are the last to form (*Towers et al., 2008*). Broadly inhibiting Fgf-signaling also shows that *Shh* operates independently of Fgf-signaling, while our cyclopamine experiments show that Shh-signaling regulates *Fgf8* expression. In addition, our finding that *Gremlin1* is regulated by Fgf-signaling suggests further rearrangement of genetic interactions observed in amniotes (*Lewandoski et al., 2000*; *Sun et al., 2002*). Together, these data show that while the role of Shh-signaling from the ZPA is conserved in salamander limb development, movement of amniote/anuran AER-Fgf ligands to the mesenchyme was accompanied by a change in how Fgf-signaling regulates limb development. Ultimately, our findings support similar molecular players being deployed during limb development across tetrapods but demonstrate that a divergent molecular program in urodeles resides predominantly in one cellular compartment: the limb mesenchyme.

While it is tempting to speculate that mesenchymal compartmentalization of limb developmental signaling is somehow causally related to regenerative ability, available data suggests otherwise. Data from basal Actinopterygians (paddlefish), Chondrichthyians (catfish) and anurans show ectodermal-mesenchymal segregation of the Shh- and Fgf-signaling (*Christen and Slack, 1998*; *Tulenko et al., 2017*). Pectoral fin regeneration is an ancient feature of Actinopterygians, Sarcopterygians and Chondrichthyians suggesting that mesenchymal core signaling alone is not exclusive to regenerating species. On the flip side, anurans show subtle molecular differences during limb development compared to amniotes, especially along the dorsoventral axis (*Christen and Slack, 1998*). Thus, it is plausible that ectodermal-mesodermal compartmentalization was the first step toward developmental canalization that ultimately increased robustness in the limb program, specifically in the autopod. Future limb development studies using a diverse array of anurans and lungfish will help shed light on these questions as will continued studies in salamanders and newts.

# Materials and methods

## Animals and tissue harvest

Axolotls (*Ambystoma mexicanum*) were acquired from the Ambystoma Genetic Stock Center (Lexington, KY) and from our own laboratory colony. Chicken eggs (University of Kentucky, Department of Animal Sciences) were incubated to stage and mouse embryos were harvested from Swiss Webster mice (ND4, Envigo, Indianapolis, IN). All procedures were conducted in accordance with, and approved by, the University of Kentucky Institutional Animal Care and Use Committee (IACUC Protocol: 2013–1174). Axolotl embryos were kept at 20–21°C and reared in glass bowls (7.5'/6'/2.5' dimensions) in 800 ml 20% Holtfreter's solution. 15–20 larvae were kept in a single bowl and were fed with brine shrimps from ~3 weeks post-fertilization. Larvae used for drug treatments, proliferation, cell death assays, qRT-PCR and DiI labeling were reared in 6-well plates. Larvae used for forelimb staging and area/snout to tail-tip measurements, Alcian blue-Alizarin red staining, in situ hybridization, histology and proliferation assays were anesthetized using 1x benzocaine (Sigma) and fixed overnight in 4% paraformaldehyde at 4°C. Larvae used for cell death assay were rinsed gently in Hanks BSS four times, five mins each on a rocker, fixed overnight in 4% PFA post-Lysotracker treatment. Developmental stages were referenced against previously reported post-hatching stages (*Nye et al., 2003*) and were extended as outlined in *Figure 1*. Individual animals (n = 35) were

examined every day to assess limb stage. Larvae used for qRT-PCR were anesthetized using 1x ben-zocaine (Sigma), limb tissue samples were snap frozen and stored at −80°C until further use. Chicken (HH25) and mouse (E11.5) embryos were harvested, fixed overnight in 4% paraformaldehyde at 4°C and processed for in situ hybridization experiments.

## Alcian blue and Alizarin red staining

Larvae were fixed overnight, washed three times, 10 mins each in PBT (Phosphate Buffer Saline and 1% Tween 20), dehydrated in graded ethanol series 25%, 50%, 75% and stored in 100% ethanol at −20°C until further use. Dehydrated larvae were washed three times, 10 mins each in 1x PBS (Phosphate Buffer Saline), stained with 0.02% Alcian blue 8GX (Sigma Aldrich) in 70% ethanol and 30% glacial acetic acid for 3 hr to overnight. Stained larvae were rehydrated in graded ethanol series (100%, 75%, 50% and water 1 hr each) and stained with 0.1% Alizarin Red (Sigma Aldrich) in 1%KOH overnight. Larvae were cleared in 1%KOH/glycerol series: 3KOH:1glycerol (imaged when cleared), 1KOH:1glycerol (1 day) and 1KOH:3glycerol (stored at room temperature).

## Histology

Larvae were fixed overnight, washed twice in 1x PBS and stored in 70% ethanol at 4°C until further use. Larvae were then processed for paraffin embedding (Histo5 Tissue Processor, Milestone) and tissue samples were sectioned at 5 µm. H and E staining was done on deparaffinized and rehydrated sections. For bright-field visualization of limb buds, Mayer's haematoxylin was used to counterstain the nuclei and coverslips were mounted with Cytoseal XYL (ThermoFisher, Waltham, MA).

## Gene isolation and riboprobe synthesis

Coding sequences for axolotl genes were obtained from NCBI, (*Bryant et al., 2017*) or www.ambystoma.org (*Supplementary file 1*). Axolotl sequences were aligned with human homologs to locate the 3'UTRs. Primers were designed against (when possible) or close to 3'UTRs of the axolotl sequences. Briefly, RNA was extracted using Trizol reagent from stage 31 larvae, stage 32 larvae or regenerating early forelimb buds from 5 to 10 cm juveniles. cDNA was synthesized from 1 to 0.5 µg RNA using SensiFast cDNA synthesis kit. Coding genes sequences were amplified out using Advantage HD polymerase kit and amplified products were ligated into pGEMT-Easy vector (Promega) via T-A cloning using manufacture's protocol. Plasmids were transformed into Max Efficiency DH5α cells (Invitrogen) and blue-white colonies were obtained. Colony PCR was done to confirm insert sequence size and positive colonies were picked for plasmid mini-prep (Qiagen). Plasmids were sent out for sequencing. Gene sequences and orientation of insertion into vector was verified and positive colonies were used for plasmid maxiprep (Zymo Research). Plasmids containing *Shh* and *Fgf8* genes for chicken and mouse were a gift from the Cohn lab, University of Florida. Plasmids containing *Keratin5* and *Keratin17* genes for axolotl were a gift from the Satoh lab, Okayama University. Plasmids containing *Fgf8* and *Gli3* genes for axolotl were a gift from the Tanaka lab, IMP, Austria. ~ 20 µg of plasmid was linearized using specific restriction enzymes to obtain the sense and antisense probe templates.

Sense and antisense probes for axolotl genes were synthesized from 2.5 µg linearized plasmids using DIG RNA Labeling Kit (SP6/T7) (Roche). Sense and antisense probes for chicken and mouse genes were synthesized from 1 µg linearized plasmids using the same kit. *In vitro* transcription of probes was carried out for 3 hr to overnight at 37°C. Probes were treated with 1unit DNAse (Promega, CAT#M610A) for 1 min at 37°C and reaction was terminated with 2 µl DNAse stop solution (Promega, CAT#M198A). Probes were purified and eluted in 50 µl of nuclease-free water using mini Quick Spin RNA Columns (Roche) or RNeasy MinElute Cleanup Kit (Qiagen), run on 1% agarose gel to access quality and quantified using NanoDrop.

## In situ hybridization experiments

Larvae/embryos fixed overnight were washed three times, 10 min each in PBT (Phosphate Buffer Saline and 1% Tween 20), dehydrated in graded methanol/PBT series 25%, 50%, 75% and stored in 100% at −20°C until further use. For all the experiments at least three larvae or embryos/stage/gene were used. Each axolotl larva was decapitated, and the bottom half of the trunk was amputated. Two axolotl larvae per stage were placed in a DNAse/RNAse free 2 ml tube and treated with 2 ml of

each solution. For chicken and mouse, one embryo was placed in a similar 2 ml tube. Dehydrated larvae/embryos were rehydrated in a graded methanol/PBT series 75%, 50%, 25%, washed with PBT twice 5 min each, bleached with 6% $H_2O_2$/1x PBS for 1 hr, washed with PBT twice 5 min each. The above steps were done under ice-cold conditions. Larvae/embryos were permeabilized with 20 µg/ml Proteinase K (Roche) in PBS for 7–10 min (40 µg/ml Proteinase K was used for *Sox9* and *Ihh* axolotl genes), washed with PBT twice for 5 min each, fixed with 0.2% Gluteraldehyde/4% paraformaldehyde and washed with PBT twice for 5 min each. The above steps were done at room temperature. Larvae/embryos were incubated overnight in hybridization buffer (5% Dextran sulphate, 2% blocking powder from Roche, 5X SSC, 0.1% TritonX, 0.1% CHAPS from Sigma Aldrich, 50% formamide, 1 mg/ml tRNA from Roche, 5 mM EDTA from Sigma and 50 µg/ml Heparin from Sigma) at 65°C. The tubes were replaced with fresh hybridization buffer, 0.1–1 µg of probe was added into each vial and incubated at 65°C for 2 days. High stringency washes were done with 2X SSC/0.1% CHAPS thrice for 20 min each, 0.2X SSC/0.1% CHAPS 4 times for 25 mins each and with KTBT (15 mM Tris-HCl pH 7.5, 150 mM NaCl, 10 mM KCl and 1% Tween 20) twice for 5 min each. Larvae were blocked with 20% goat serum in KTBT for 3 hr. Later, fresh blocking solution was added. An anti-Digoxigenin-AP, Fab fragment antibody (Roche) was added at 1:3000 dilution and incubated overnight at 4°C. Larvae were washed with KTBT 5 times for 1 hr each and then incubated in KTBT overnight at 4°C. Larvae were washed with NTMT (100 mM Tris-HCl pH 9.5, 50 mM $MgCl_2$, 100 mM NaCl and 1% Tween 20) twice for 5 min each and incubated in NBT/BCIP (Roche) solution in NTMT (BCIP-0.17 mg/ml, NBT-0.33 mg/ml, 10% DMF) till a signal developed with minimum background staining. Limbs were photographed and larvae were washed with TE buffer (10 mM Tris HCl, pH 8 and 1 mM EDTA, pH eight made up in DEPC treated water) 3 times for 10 mins each and fixed in 4% PFA until further use.

## Cryosectioning post in situ hybridization experiments

Larvae from *Keratin5* and *Keratin17* in situ hybridizations were washed in TE buffer (10 mM Tris HCl pH 8 and 1 mM EDTA pH 8) 3 times for 10 min each, fixed in 4% PFA for 1 hr and washed with 1x PBS 3 times for 5 min each. Larvae were transferred into 2 ml vials with 30% sucrose in 1x PBS and placed on a rotor for 20 mins till they sank to the bottom. Larvae were then placed in OCT for 25 min and frozen for cryo-sections. Cryo-sections were taken at 5 µm thickness, dried overnight at 37°C, fixed with 4% PFA for 10 min, washed with 1x PBS 3 times for 5 min each, treated with Hoechst solution (1:10,000 dilution), air dried, sealed with ProLong Gold antifade reagent (Invitrogen) and imaged.

## qRT-PCR analysis

For drug experiments, post-hatch axolotl larvae were reared in 6-well plates in either 1.5% DMSO, 45 µM SU5402, 0.02% ethanol or 1 µg/ml cyclopamine till stage 46 (see below or treatment details). Whole limbs were dissected from the body wall, immediately snap frozen and stored at −80°C until RNA extraction. Three replicates were used for each condition (treatment/control) and each replicate represented a pool of limbs (both left and right) from 10 to 20 animals.

To validate the anterior versus the posterior expression of *Shh*, *Fgf8* and *Gli3* genes, larvae were reared in glass bowls (7.5'/6'/2.5' dimensions) in 800 ml 20% Holtfreter's solution. Whole limbs were dissected from the body wall, further dissected into anterior and posterior halves/compartments (*Figure 2—figure supplement 3D*), immediately snap frozen and stored at −80°C until RNA extraction. n = 2 or 3 was used and each set was pooled from 20 limbs (both right and left).

RNA was extracted using Trizol reagent (Invitrogen). cDNA was synthesized from 1 to 0.5 µg RNA using SensiFast cDNA synthesis kit. The following primers were used for qRT-PCR:

> *Shh* Forward- ATTTTTAAGGACGAAGAGAACACCG,
> *Shh* Reverse- CTTATCCTTACACCTCTGGGTCATT,
> *Fgf8* Forward- ATTAATTGTGGAAACGGACACCTTC,
> *Fgf8* Reverse- AATCAGCTTTCCCTTCTTGTTCATG,
> *Gli3* Forward- CATGGATGTGGTCGTTATTGATGTG,
> *Gli3* Reverse- GAGGTTATTTACGAGACCGACTGTC,
> *Etv1* Forward- TCTTGGAAGAGTTCTTCTGAGTCAT
> *Etv1* Reverse- CGTGTGAGAAATTGTAACGAGAGA
> *Etv4* Forward- ACTATGCATACGATTCAGATGTTCC
> *Etv4* Reverse- ATAGCCCTCCACGTTCATATACATT

*Gremlin1* Forward- GGACACCCAGAATACTGAGCA
*Gremlin1* Reverse- GTAGACCAATCGAAACATCCTGT
*Ptch1* Forward- TGTAGATCTGCTCCAATGCAAAC
*Ptch1* Reverse- CTGACCCGGAGTACTTGCAG
*Tubulin alpha* Forward- CCCAGGGCCGTGTTCGTC
*Tubulin alpha* Reverse- CCGCGGGCGTAGTTGTTG
*GAPDH* Forward- AAAAGGTCTCCTCTGGCTATGAC
*GAPDH* Reverse- AGGGCTATAAAAGAGCATTATCGAG
*RLP32* Forward- AGGCTACTGGGAGTTTTAATAAGGA
*RLP32* Reverse- AGATTACAGCACCCACTGTCTTTT

*Tubulin alpha* was used as the internal control/house-keeping gene for limbs obtained from larvae reared in 20% Holtfreter's solution. *GAPDH* and *RLP32* were used as the internal control/house-keeping genes for the DMSO/SU5402 and ethanol/cyclopamine experiments, respectively, since there was no significant fold change in the $2^{-Ct}$ values (*Schmittgen and Livak, 2008*) (*Supplementary file 2*). $2^{-\Delta\Delta Ct}$ method was used to calculate the fold change values between control (DMSO or ethanol) and treatment (SU5402 or cyclopamine) groups (*Schmittgen and Livak, 2008*).

## DiI labeling of the proximal *Shh* domain

Post-hatch axolotl larvae were reared in 6-well plates in 3 ml Holtfreter's solution. DiI (D-282, molecular probes) was dissolved in dimethyl formamide at 3 mg/ml concentration. Larvae between stages 45 and 46 were anesthetized in 1x benzocaine and approximately 5 nl (0.05–0.20 mm diameter) of DiI was injected into the approximate position of the posterior *Shh* domain. A total of 34 limbs were analyzed for this experiment. Images were taken immediately to confirm the domain specific restriction of the injection and fluorescence was tracked every 2 days till all four digits formed completely. All images were taken post-anesthesia in 1x benzocaine.

## Analysis of single-cell RNA-seq data

Single-cell RNA-seq data from embryonic limb buds (*Gerber et al., 2018*, Table S7) were analyzed as follows: Gene expression (log2(TPM)) from stages 40 and 44 (as reported in Gerber), and which correspond to our stages 44 and 45 (Prayag Murawala, personal communication) were used as input for principal component analysis. The top twenty loadings of the positive and negative axes of the first five principal components were inspected to identify principal components that segregated developmental axes markers on separate ends. No reliable anterior-posterior, or dorsal-ventral markers segregated in this manner in the current dataset. However, proximal-distal markers (Meis2 and Hoxd11) were segregated on opposite ends of principle component three in the PCA of cells from stage 44, but not 40. In order to orient PC3 so that Hoxd11 expression is on the right and Meis2 on the left (i.e., conventional proximal-distal orientation), PC3 values were multiplied by -1. To determine co-expression of *Fgf* ligands and receptors, expression was defined as the presence of one or more transcript per million.

## SU5402 and cyclopamine treatments

Axolotl larvae at pre-limb bud stage 39 were reared in 6-well plates and treated with 3 ml solutions for all drug experiments. A working stock of 3 mM SU5402 (Sigma) was made in DMSO and then diluted to 45 µM in 3 ml 20% Holtfreter's solution per well. Control larvae were treated with an equivalent amount of DMSO (1.5%) in 20% Holtfreter's solution. Larvae were kept in the dark and the solution was changed every three days. At three weeks post fertilization each larva was fed 20–30 brine shrimp every day.

Cyclopamine treatments were performed as previously described (*Stopper and Wagner, 2007*). A working stock of 5 mg/ml of cyclopamine was made in 100% ethanol and 0.6 µl from this stock was added into 3 ml 20% Holtfreter's solution per well (1 µg /ml final concentration). An equal amount of 100% ethanol (0.02%) was added into control wells. Larvae were kept in dark and solutions were replenished every two days. At three weeks post fertilization each larva was fed 20–30 brine shrimps every day.

## Limb area and snout to tail measurements

Limb area was measured for stage 45 and stage 46 larvae that were reared in 1.5% DMSO, 45 µM SU5402, 0.02% ethanol or 1 µg/ml cyclopamine. Snout to tail-tip lengths were measured for stage 46 larvae that were reared in 1.5% DMSO, 45 µM SU5402, 0.02% ethanol or 1 µg/ml cyclopamine. All measurements were made using Fiji software (NIH) after calibrations.

## Cell proliferation assay using EdU labeling

Post-hatch larvae were reared in 6-well plates in 3 ml of either of the solutions: 20% Holtfreter's solution, 1.5% DMSO, 45 µM SU5402, 0.02% ethanol or 1 µg/ml cyclopamine. Larvae were additionally treated with 0.1 mg/ml of EdU at stage 46 for 24 hr, fixed overnight in 4% PFA, washed with 1x PBS twice for 5 min, dehydrated in 1x PBS/methanol series (25%, 50%, 75% and 100% methanol, 5 min each) and stored in 100% methanol at −20°C until further use.

For EdU staining, all steps were performed at room temperature unless mentioned otherwise. Larvae were rehydrated backwards through the methanol series starting at 100% methanol and ending at 100% 1x PBS. This was followed by PBT washes for 5 min twice and 2.5% trypsin (Gibco) treatment for 10 min. The larval limbs were checked for clarity at this point. Larvae were washed with water for 5 min twice, treated with 20 µg/ml of proteinase K in PBT for 7–10 min, washed with water for 5 min twice, fixed in 100% acetone at −20°C for 10 min, washed with water for 5 min once, washed with PBT for 5 min once, incubated in fresh click reaction solution (1x TRIS buffer saline, 4 mM $CuSO_4$ in 1x TRIS buffer saline, 2 µl Alexa-flour-594 Azide (Life technologies), 1 mM sodium ascorbate in 1x TRIS buffer saline) for 30 min on a rocker in the dark, washed with 1x PSB for 5 min thrice, incubated in DAPI (1:1000 dilution) for 30 mins, washed with 1x PSB for 5 min thrice, checked for fluorescence under a stereomicroscope and stored at 4°C in the dark till lightsheet imaging.

## Cell death assay using LysoTracker

Post-hatch larvae were reared in 6-well plates in 3 ml of either of the solutions: 20% Holtfreter's solution, 1.5% DMSO, 45 µM SU5402, 0.02% ethanol or 1 µg /ml cyclopamine. Larvae were transferred into 24-well plates and treated with 200 µl of 5 µM LysoTracker Red DND-99 (molecular probes) in Hanks BSS for 45 min to 1 hr at 20–21°C. Larvae were rinsed gently in Hanks BSS four times, 5 min each on a rocker, fixed overnight in 4% PFA, rinsed once in Hanks BSS for 10 min and dehydrated through a methanol series in Hanks BSS (50%, 75%, 80%, 100%, 5 min each step) to eliminate background staining and stored in 100% methanol at −20°C until imaging.

## Microscopy and image analysis

Whole mount images for limb staging/size measurements, Alcian blue-Alizarin red staining, in situ hybridization, DiI experiments and apoptosis assays were taken on an SZX10 light microscope (Olympus, Tokyo, Japan) using a DP73 CCD camera (Olympus). *Figure 1—figure supplement 1* depicts all different forelimb stages to scale. Forelimb images are presented unscaled for in situ hybridizations except where treatment/control limbs are presented. Images for H and E staining and cryo-sections (from *Keratin5* and *Keratin17* wholemount hybridizations) were taken on a BX53 microscope (Olympus, Tokyo, Japan) with DP80 CCD camera. Both the microscopes were equipped with CellSense software (CellSense V 1.12, Olympus corporation).

EdU stained stage 46 larvae were imaged using a Zeiss Lightsheet Z.1 (College of Arts and Science Imaging Centre, University of Kentucky). Larvae were embedded in 1% low melting agarose (Sigma) dissolved in 1x PBS, mounted in a glass capillary (2.15 mm inner diameter) and the glass capillary was placed in a chamber (specific for the 20x objective lens) filled with 1x PBS. Imaging was done with Zen software (Zeiss) and samples were excited using 561 nm and 488 nm lasers. 20x objective was used and images of all the limbs were taken at 0.75 zoom. Either right, left or both lightsheets were used based on the orientation of the limbs. Image processing and analysis was done with Arivis vision4D software (Arivis). The czi extension files from Zen software was imported into the Arivis vision4D software. An object mask was hand drawn at each z-planes based on the DAPI signal to outline the limb and create a limb mask for total limb volume calculations. The following pipeline was made to calculate red cell aggregate volume and total limb volume: Input ROI (current image set, all channels, scaling 50%), object mask (add the hand drawn object), denoising filter (mean, radius = 10), Intensity filter (radius = 9, K = 0, offset = −10, binary - false), denoising filter

(median, radius = 10) and result storage (intensity threshold, range specified, allow holes while quantification). Volume values in $\mu m^3$ and voxel counts were given as outputs.

## Statistical analyses

All statistical analyses were made using JMP (version Pro 12.10, SAS Institute Inc). Box plots for forelimb staging were made using graph builder in JMP. The vertical line within the box represents the median number of larvae found at a specific stage and days post fertilization. The ends of the box represent the 25th and 75th quartiles and the whiskers on either side represent the interquartile range. For qRT-PCR data the $2^{-\Delta\Delta Ct}$ method was used to calculate fold changes of genes between control (DMSO or ethanol) and treatment (SU5402 or cyclopamine) groups. Calculations for mean Ct values, $\Delta Ct$ values for experimental and control groups, $\Delta\Delta Ct$ values between experimental and control groups, $2^{-Ct}$ and $2^{-\Delta\Delta Ct}$ fold change values were made using Microsoft Excel and detailed in *Supplementary file 2*. Student's t-test was used to calculate the significant changes in relative gene expressions between control and treatment groups. For limb size and snout to tail-tip measurements, student's t-test was used to calculate the significant changes between control and treatment groups. Differences were considered significant if p<0.05. To analyze whether the decrease in limb size post drug treatment was due to the decrease in overall body length (snout to tail-tip length) we used a fit model ANOVA with treatment, overall body length and treatment*overall body length interaction.

To evaluate co-expression of *Fgf* ligands and receptors along the limb proximal-distal axis from single-cell RNAseq data, we utilized the fact that contribution to principle component three followed a normal distribution according to a Shapiro-Wilks normality test (W = 0.95551, p-value<2.2e-16). A Welch Two Sample t-test test was conducted to test the hypothesis that two populations (*Fgf* ligands and receptors) have the same mean contribution to principle component 3.

For lightsheet data, red cell aggregate volume/limb volume (%) was calculated in Microsoft Excel. Comparisons within control and treatment groups were determined by one-way ANOVA (Kruskal-Wallis tests).

## Specific statistical information

| Experiment/figure number | Statistics |
|---|---|
| *Figure 5B*: Limb size for stage 46 DMSO or SU5402 treatments | Unpaired one-tailed Student's t-test<br>• n = 5 (DMSO), n = 6 (SU5402)<br>• Mean (DMSO) = 0.1614, Mean (SU5402) = 0.072<br>• SD (DMSO) = 0.015582, SD (SU5402) = 0.0177539<br>• SEM (DMSO) = 0.006969, SEM (SU5402) = 0.007248<br>• p-value<0.0001<br>• T = −8.775<br>• 95% confidence interval = −0.06636 to −0.11244 |
| *Figure 5E*: qRT-PCR data for DMSO or SU5402 treated stage 46 limbs | Unpaired two-tailed Student's t-test<br>• n = 3 (DMSO), n = 3 (SU5402)<br>• Mean (DMSO) = 1.09 (*Etv1*), 1 (*Etv4*), 0.97 (*Shh*), 0.85 (*Ptch1*), 1.1 (*Gremlin1*)<br>• Mean (SU5402) = 0.28 (*Etv1*), 0.4 (*Etv4*), 0.37 (*Shh*), 1.25 (*Ptch1*), 0.47 (*Gremlin1*)<br>• SD (DMSO) = 0.055 (*Etv1*), 0.1 (*Etv4*), 0.06 (*Shh*), 0.18 (*Ptch1*), 0.07 (*Gremlin1*)<br>• SD (SU5402) = 0.05 (*Etv1*),. 04 (*Etv4*), 0.12 (*Shh*), 0.22 (*Ptch1*), 0.24 (*Gremlin1*)<br>• SEM (DMSO) = 0.03 (*Etv1*),. 06 (*Etv4*), 0.03 (*Shh*), 0.11 (*Ptch1*), 0.04 (*Gremlin1*)<br>• SEM (SU5402) = 0.03 (*Etv1*),. 02 (*Etv4*), 0.08 (*Shh*), 0.13 (*Ptch1*), 0.14 (*Gremlin1*)<br>• p-value=<0.0001 (*Etv1*), 0.0006 (*Etv4*), 0.0022 (*Shh*), 0.07 (*Ptch1*), 0.014 (*Gremlin1*)<br>• T = −19.13 (*Etv1*), −9.79 (*Etv4*), −6.95 (*Shh*), 2.44 (*Ptch1*), −4.14 (*Gremlin1*)<br>• 95% confidence interval = −0.7 to −0.91 (*Etv1*), −0.43 to −0.77 (*Etv4*), −0.36 to −0.83 (*Shh*), 0.9 to −0.05 (*Ptch1*), −0.2 to −1 (*Gremlin1*) |

*Continued on next page*

Continued

| Experiment/figure number | Statistics |
| --- | --- |
| **Figure 5I**: Cell proliferation in DMSO or SU5402 treated stage 46 limbs | One-way ANOVA (Kruskal-Wallis tests)<br>• n = 3 each for DMSO and SU5402<br>• Median: DMSO = 29.15, SU5402 = 4.75<br>• p-value=0.0495 |
| **Figure 5—figure supplement 1A and C**: Limb size for stage 45 DMSO or SU5402 and ethanol or cyclopamine treatments | Unpaired one-tailed Student's t-test<br>• n = 15 (DMSO), n = 15 (SU5402)<br>• Mean (DMSO) = 0.082, Mean (SU5402) = 0.074<br>• SD (DMSO) = 0.016, SD (SU5402) = 0.012<br>• SEM (DMSO) = 0.004, SEM (SU5402) = 0.003<br>• p-value=0.0514<br>• $T = -1.68637$<br>• 95% confidence interval = −0.019 to 0.002<br>• n = 6 (ethanol), n = 6 (cyclopamine)<br>• Mean (ethanol) = 0.071, Mean (cyclopamine) = 0.063<br>• SD (ethanol) = 0.012, SD (cyclopamine) = 0.006<br>• SEM (ethanol) = 0.002, SEM (cyclopamine) = 0.005<br>• p-value=0.909<br>• $T = 1.43$<br>• 95% confidence interval = −0.004 to 0.02 |
| **Figure 5—figure supplement 1B and D**: Snout to tail-tip length for DMSO or SU5402 and ethanol or cyclopamine treatments | Unpaired one-tailed Student's t-test<br>• n = 5 (DMSO), n = 6 (SU5402)<br>• Mean (DMSO) = 10.92, Mean (SU5402) = 10.29<br>• SD (DMSO) = 0.2, SD (SU5402) = 0.25<br>• SEM (DMSO) = 0.088, SEM (SU5402) = 0.104<br>• p-value<0.0007<br>• $T = -4.5$<br>• 95% confidence interval = −0.316 to −0.950<br>• n = 6 (ethanol), n = 6 (cyclopamine)<br>• Mean (ethanol) = 10.5, Mean (cyclopamine) = 9.98<br>• SD (ethanol) = 0.15, SD (cyclopamine) = 0.3<br>• SEM (ethanol) = 0.06, SEM (cyclopamine) = 0.121<br>• p-value<0.0016<br>• $T = 3.87$<br>• 95% confidence interval = 0.82 to 0.222 |
| **Figure 6B**: Limb size for stage 46 ethanol or cyclopamine treatments | Unpaired one-tailed Student's t-test<br>• n = 6 (ethanol), n = 6 (cyclopamine)<br>• Mean (ethanol) = 0.16, Mean (cyclopamine) = 0.095<br>• SD (ethanol) = 0.008, SD (cyclopamine) = 0.016<br>• SEM (ethanol) = 0.003, SEM (cyclopamine) = 0.006<br>• p-value<0.0001<br>• $T = 8.36$<br>• 95% confidence interval = 0.08 to 0.045 |
| **Figure 6C**: qRT-PCR data for ethanol or cyclopamine treated stage 46 limbs | Unpaired one-tailed Student's t-test<br>• n = 3 (ethanol), n = 3 (cyclopamine)<br>• Mean (ethanol) = 1.12 (*Ptch1*), 0.94 (*Gremlin1*), 1.12 (*Fgf8*)<br>• Mean (cyclopamine) = 0.7 (*Ptch1*), 0.39 (*Gremlin1*), 0.22 (*Fgf8*)<br>• SD (ethanol) = 0.145 (*Ptch1*), 0.12 (*Gremlin1*), 0.14 (*Fgf8*)<br>• SD (cyclopamine) = 0.00004 (*Ptch1*), 0.1 (*Gremlin1*), 0.03 (*Fgf8*)<br>• SEM (ethanol) = 0.08 (*Ptch1*), 0.07 (*Gremlin1*), 0.08 (*Fgf8*)<br>• SEM (cyclopamine) = 0.02 (*Ptch1*), 0.06 (*Gremlin1*), 0.02 (*Fgf8*)<br>• p-value=0.0039 (*Ptch1*), 0.002 (*Gremlin1*), 0.0002 (*Fgf8*)<br>• $T = 4.9$ (*Ptch1*), 5.97 (*Gremlin1*), 11.21 (*Fgf8*)<br>• 95% confidence interval = 0.66 to 0.19 (*Ptch1*), 0.8 to 0.3 (*Gremlin1*), 1.1 to 0.68 (*Fgf8*) |
| **Figure 6G**: Cell proliferation in ethanol or cyclopmaine treated stage 46 limbs | One-way ANOVA (Kruskal-Wallis tests)<br>• n = 3 each for ethanol and cyclopamine<br>• Median: ethanol = 30.06, cyclopamine = 13.9,<br>• p-value=0.0495 |

## Acknowledgements

The authors thank Ryan Woodcock for help with axolotl bioinformatics, Elly Tanaka and Ina Stutzer for access to genomic data, James Monaghan and Malcolm Maden for discussion, Jakub Famulski and Douglas Harrison for assistance with lightsheet imaging and image processing. We are grateful to our anonymous reviewers for their extremely helpful comments.

## Additional information

### Funding

| Funder | Grant reference number | Author |
|---|---|---|
| National Science Foundation | IOS -1353713 | Ashley W Seifert |

The funders had no role in study design, data collection and interpretation, or the decision to submit the work for publication.

### Author contributions

Sruthi Purushothaman, Conceptualization, Formal analysis, Validation, Investigation, Visualization, Methodology, Writing—original draft, Writing—review and editing; Ahmed Elewa, Formal analysis, Investigation, Methodology, Writing—review and editing; Ashley W Seifert, Conceptualization, Resources, Formal analysis, Supervision, Funding acquisition, Investigation, Methodology, Writing— original draft, Project administration, Writing—review and editing

### Author ORCIDs

Sruthi Purushothaman (iD) https://orcid.org/0000-0002-3974-4731
Ahmed Elewa (iD) http://orcid.org/0000-0002-1988-7970
Ashley W Seifert (iD) https://orcid.org/0000-0001-6576-3664

### Ethics

Animal experimentation: All procedures were conducted in accordance with, and approved by, the University of Kentucky Institutional Animal Care and Use Committee (IACUC Protocol: 2013-1174).

### Decision letter and Author response

Decision letter https://doi.org/10.7554/eLife.48507.sa1
Author response https://doi.org/10.7554/eLife.48507.sa2

## Additional files

### Supplementary files

• Supplementary file 1. Details of gene sequences used to make the antisense probes for in situ hybridization experiments. The excel file contains gene names, source/NCBI accession id, full length/ partial coding sequences and ids, human hit, e-value for human hit, forward primer sequence, reverse primer sequence and length of antisense probe used.

• Supplementary file 2. Gene expression analysis for qRT-PCR using excel. The excel file contains gene expression analysis for *Figure 2—figure supplement 3D* and DMSO/SU5402 and ethanol/ cyclopamine drug experiments.

• Transparent reporting form

### Data availability

Sequencing data have been previously published and are publicly available. Source data have been provided in Supplementary files 1 and 2.

The following previously published dataset was used:

| Author(s) | Year | Dataset title | Dataset URL | Database and Identifier |
|---|---|---|---|---|
| Gerber T, Murawala P, Knapp D, Masselink W, Schuez M, Hermann S, Gac-Santel M, Nowoshilow S, | 2018 | scRNAseq dataset, Table S7 | https://www.ncbi.nlm. nih.gov/geo/query/acc. cgi?acc=GSE106269 | NCBI Gene Expression Omnibus, GSE106269 |

Khattak S, Currie JD, Camp JG, Tanaka EM, Treutlein B

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
