## [Decision Letter]

Thank you for submitting your work entitled "*Shh* mediated *FGF*-signaling is compartmentalized in the mesenchyme and controls limb size during salamander development" for consideration by *eLife*. Your article has been reviewed by a Senior Editor, a Reviewing Editor, and two reviewers. The reviewers have opted to remain anonymous.

Our decision has been reached after consultation between the reviewers. Based on these discussions and the individual reviews below, we regret to inform you that your work will not be considered further for publication in *eLife* in its current form.

As you can see below, both reviewers agree that the work is of potentially great importance. In our discussions, we all agreed that your work is an ambitious effort to examine in axolotls at least 4 aspects of limb development (AER signaling, ZPA signaling, dorsoventral patterning, and cartilage differentiation) that have each been studied for decades in chick and mouse. The data presented indicate a number of major differences in the mode and mechanisms of limb formation in salamander relative to chick, mouse, and other research organisms, challenging a number of current paradigms of limb development. Our problem with these findings is that the conclusions are overly bold given the relative superficiality of the analyses. For example, our biggest concern is that only Fgf8 was looked at when it has been shown by many that multiple Fgfs are involved in AER signaling. Because the argument about loss of ectodermal FGF signaling is central to the paper, it strikes us as essential to look at expression of the Fgfs that are involved in AER signaling (Fgf4/8/9/17) at minimum, and probably to look at distribution of the 4 FGF receptors., so they need to be thorough in their analysis.

We appreciate that addressing the points above is a significant amount of work. If you choose to address these and the other comments raised below and submit a more in-depth manuscript later on, we would be happy to re-examine the work. If, on the other hand, you would prefer to have the work published elsewhere as is we are enclosing the full text of the reviews, which we hope you will find useful. As you will see, both reviewers concurred on many aspects. To facilitate your evaluation of the enclosed reviews, we have identified those comments which are essentially the same. I will use R1 and R2 to identify the reviewers, followed by point numbers of the respective reviews.

• R1-1 = R2-9;

• R1-2 = R2-5 and R2-6;

• R1-4 = R2-4 and part of R2-3;

• Questions about the SU5402 treatments are discussed in R15, R2-3 and R2-14;

• Different but complementary questions about Shh expression are raised in R1-6 and R2-1;

• R1-7 = R2-2

*Reviewer #1:*

This study makes the bold claim that Urodeles, as exemplified by the axolotl Ambystoma mexicanum, exhibit distinct developmental features compared to other tetrapod species. An intriguing implication of this claim is that a difference in developmental program may be related to the ability of axolotls to regenerate limbs. The study covers a wide variety of features: (1) The morphology of the limb bud, including whether or not an AER forms, (2) the order of deposition of an Alcian blue positive matrix and the expression of Sox 9, Ihh; (3) order of digit formation, (4) Proximal distal patterning (via Fgf8 expression, SU5402 treatment) (5) A/P specification (via Shh expression, cyclopamine treatment) (6) Relationship between the P/D and A/P regulatory loops (via Grem. Ptch1 regulation), and (7) D/V patterning (via Wnt, En1 expression etc).

Unfortunately, the study handles each of these issues very superficially considering that there are at least 100 major studies examining each of these topics using chicken or mouse model systems over the course of the last 25 years. As a consequence, this reviewer believes that the bold conclusions drawn based on the data collected so far, while intriguing, are much too premature.

Essential revisions:

1) Just because there is no distal pseudo-stratified epithelial thickening (AER) doesn't mean an ectodermal signaling center is not there. There are other examples of amphibians that have no AER and still have AER-FGF expression.

2) Typically condensation is defined by histological coalescing of cells, however this study does not use histology to determine the order of digit condensation. If condensation is defined by proxies for cartilage differentiation the authors need to explain the variation in Alcian blue intensity for digits (order = 2?1? or vice versa, not clear from images 3, 4), sox 9 (2,1, 3, 4), and Ihh (1, 2? 3, 4). The authors should compare their more careful findings with those of Sue Mackem and colleagues (PMID: 18410737 and 21509901). Axolotls have 4 digits, is it digit 1 that is missing or digit 5?

3) The authors should be aware that colormetric RNA in situ hybridization is not as sensitive as other RNA detection methods (FISH or RNAscope). Thus, lack of color deposition, especially using whole mount preps, does not necessarily mean that no expression is present. For example, it may be the case that there is an early distal swath of *Sox9* expression but that it present at too low levels to detect give the method used.

4) There are at least 22 Fgfs and 4 FGF receptors. In many comparisons between species, related growth factors have been shown to substitute for each other. For example, Sox10 appears to provide the function of mouse *Sox9* in zebrafish. The expression pattern for Fgf8 shown in this study is reminiscent of the pattern of Fgf10 expression in chicken/mouse embryos. Without examining at least a handful of the most well-known FGF genes (Fgf4, Fgf10, *Fgf9*, Fgf2, *Fgfr1, Fgfr2*), it is premature to claim that axolotl limbs develop without needing an ectodermal source of FGFs.

5) SU5402 treatments typically result in a mild phenotype but based on their table of results CAN result in more severe digit loss phenotypes. This suggests that either the dose or timing of administration can be modulated to achieve phenotypes that match mouse KO phenotypes.

(Perhaps best compared to *Fgfr1*/2 combo KOs). In addition, given the long period between establishment of the limb field and limb outgrowth, it may be that one would need to administer SU5402 much earlier (prior to limb bud outgrowth) to inhibit proximal specification in these species.

6) The observation that Shh is also expressed in an anterior domain is interesting and should be examined. Given that cyclopamine administration gives the expected results, is this anterior domain not significant? Does the location of Shh expression correlate with areas of cell death at the anterior and posterior autopod margins?

7) The authors make the major claim that the FGF and Hh regulatory loops are not interlinked. This is based on examining a few read-outs by whole mount RNA in situ hybridization and is not comprehensively or quantitatively assessed. Furthermore, analysis of cell death is not presented, nor are genes or proliferation rates examined PRIOR to differences in limb bud size being evident.

8) Finally the analysis of D/V patterning genes seems tacked on and could be eliminated from the paper.

In its current state this reviewer recommends rejection of the paper as a thorough examination of the above issues would certainly take longer than the 2-3 months typically allotted for *eLife* submissions. Should the bold claims hold after more detailed analysis, (suggest focusing on FGF and HH pathways as most interesting and critical, while describing the order of condensation/differentiation is likely less revelatory), the study could be of great interest and significance.

*Reviewer #2:*

Summary:

In this fascinating paper, Purushothaman and Seifert examine limb development in the axolotl, a salamander that has long been the primary model for studies limb regeneration but has been largely overlooked for developmental studies. The lack of data on how axolotl limbs develop, compared to the developmental workhorses of chick and mouse, is a major problem because the data on axolotl limb regeneration are usually compared/contrasted with data on chick and mouse limb development, with an underlying assumption that limb development is similar across these species. Thus, the importance of the data presented here, which identifies a number of major differences in the mode and mechanisms of limb formation in salamander relative to chick, mouse, and other model and non-model organisms, is difficult to overstate. Their results not only highlight the manner in which developmental mechanisms have evolved in salamanders, but they also challenge some current paradigms of limb development that are based on work in chick and mouse, and they provide a developmental baseline for comparison with regeneration studies in the same species.

The authors begin by examining skeletal development and they validate some earlier work that showed differences in the spatiotemporal pattern of digit formation. They confirm the earlier observation of Sturdee and Connock, (1975) that the axolotl limb bud lacks an AER, and they go on to show that, unlike several other tetrapods that have been reported to lack an AER, axolotls show no ectodermal expression of Fgf8. This contrasts with the results of Han et al., (2001), who reported that Fgf8 is expressed in both the ectoderm and mesenchyme. Purushothaman and Seifert go to great lengths to detect Fgf8 in both tissues and they show, unequivocally in my opinion, that the Fgf8 expression domain is entirely mesenchymal.

Based on current models of limb development from mouse and chick, multiple FGFs (from the AER) positively regulate Shh, which then signals to the AER (directly) to maintain FGF expression, and to the limb mesenchyme, where it induces Gremlin1 expression. Mesenchymal Gremlin1 antagonizes Bmps in order to maintain FGF expression in the AER. The authors report that in axolotl limb buds, Shh expression was unaffected by suppression of FGF signaling using an FGF receptor antagonist, SU5402, which is a surprising finding because chick/mouse limb experiments would predict that near-complete suppression of FGF signaling would lead to reduced Shh expression. This result is made even more interesting by their discovery that Grem1 expression is virtually eliminated in SU5402-treated limbs. When Hedgehog signaling is blocked by cyclopamine treatment, Gremlin1 and Fgf8 expression is reduced, and the limb buds are smaller. The authors conclude that, in axolotl limb buds, FGF signaling is not required for Shh expression but is required for Gremlin expression and limb size, and that SHH controls distal limb patterning and growth by regulating Grem1 and Fgf8.

Finally, they turn to dorsoventral patterning and show that Wnt7a/Radical fringe and Engrailed1, which mark, respectively, the dorsal and ventral ectoderm in chick and mouse limbs, are expressed throughout the limb mesenchyme. This is another major difference between axolotl and the major model systems, and opens many new questions about the roles of those genes in dorsovental patterning of limb structures. Although not discussed, loss of dorsoventrally polarized expression also fits with loss of the AER. Polarized expression of these genes in limb ectoderm establishes the AER at the tip of the limb bud in chick and mouse. Furthermore, chick "Wingless" mutants form and then lose the AER, after which the dorsoventral markers lose their restricted expression (Ohuchi et al., 1997). It would be worth discussing whether the authors think that the dorsoventral change is a cause or consequence of the loss of an AER.

They conclude that the mesenchymal-epithelial (Shh-Grem-Fgf8) feedback loop that controls limb development in other tetrapods has been re-arranged in axolotls, where it is entirely mesenchymal and has evolved different regulatory interactions. In a rich but short discussion, the authors explore the implications of these findings for understanding genetic interactions in the limb, evolution of developmental mechanisms, and molecular control of regeneration. The brevity of the discussion does a disservice to the many sacred cows that this paper has toppled, and I would encourage them to break apart the results and discussion so that a proper airing of these ideas can be presented. Overall, the data are clear and convincing and the discoveries are likely to be of interest to a broad group of biologists in the areas of development, regeneration and evolution.

Essential revisions:

1) The Shh expression pattern is quite interesting and makes one whether the cells that form the digits ever see the Shh signal. Although the initial domain in the posterior part of the limb is typical of early limb buds of tetrapods and fishes, the subsequent domains are very different from the patterns reported for other tetrapods. They state, "Posterior expression is never detected in the autopod" (Figure 2 legend). If the digits progenitors arise from (or near) the Shh domain, then the role of Shh in digit development could be conserved. If the cells of the Shh domain are distant from the digit progenitors, then this would challenge the current paradigm of digit development. This could be tested quite easily by labeling the initial Shh domain (at stage 45) with an injection (or carefully applied crystals) of an indocarbocyanine dye, like DiI, and then examining the distribution of labeled cells several stages later in order to determine whether those cells contribute to the digits. Determining whether digit progenitors reside within the Shh domain would allow the authors to make a direct comparison with the Shh-expressing (ZPA) cells of mouse and chick limb buds, which contribute to posterior digits.

2) Because the SU5402 treatments show that FGF signaling does not appear to maintain Shh in axolotls, is there really is a FGF-Shh feedback loop in axolotl limb bud mesenchyme?

3) Do the authors think that the same mesenchyme cells that are expressing Fgf8 are also responding to Fgf8? Do they express an appropriate receptor? How does the topological shift in the position of the ligand, Fgf8, relate to the ability of SU5402, a FGF receptor antagonist, to produce such different outcomes in axolotl vs chick and mouse limbs? Do the authors think that there is combinatorial FGF signaling in axolotls (as there is in chick and mouse), but the ligands are produced and received by the mesenchyme? If so, then can they suggest an explanation for how the same Fgfs could regulate Shh when secreted by the AER(as in chick/mouse) but do not regulate Shh when produced by the mesenchyme? Chick experiments in which the AER is removed and FGF beads are implanted into the limb bud mesenchyme suggests that the source of the ligand is irrelevant. Have they considered the possibility that Fgf8 acquired a novel function during axolotl limb evolution and lost its roles as a Shh regulator and an outgrowth signal? The proliferation data suggest that at least some of these roles are conserved.

4) In several places, the Fgf8 result is generalized to the other Fgfs involved in AER signaling (e.g., Results and Discussion section: "Taken together, our expression data clearly shows that the Shh-Grem-FGF feedback loop is entirely expressed within the mesenchymal compartment during salamander limb development."). In chick and mouse limbs, at least 4 Fgfs (Fgf4, Fgf8, *Fgf9*, and Fgf17) are expressed in the AER and signal through mesenchymal FGF receptors. Whilst it would be informative to examine the expression patterns of these other Fgfs in axolotl limb buds to determine whether the Shh-FGF feedback loop is entirely in the mesenchyme (or if some of the Fgfs have remained in the epithelium) if they choose not to examine the expression of these other Fgfs, then they should tailor the discussion accordingly and at least mention the alternate possibilities. For example, they show that Fgf8 is expressed in the mesenchyme, but it is unknown whether or where other Fgfs are expressed in the limb. Thus, several scenarios are possible, including (a) Fgf8 is mesenchymal but Fgf4, *Fgf9*, and Fgf17 are expressed in the epithelium, (b) All 4 Fgfs are mesenchymal, (c) Fgf8 is mesenchymal and the other 3 Fgfs are not expressed. Without knowing which of these scenarios is true, one must be cautious and clear with the interpretation the SU5402 results since SU5402 treatment probably suppresses all FGF signaling. For example, scenarios (b) and (c) would support "a topological rearrangement of the Shh-Grem1-FGF loop", but scenario (a) would not because 3 of the 4 Fgfs would be expressed in ectoderm. To be clear, I am not insisting that the authors examine every FGF and FGF receptor in axolotl limbs, but they need to consider the complexity of FGF signaling in amniotes in their interpretation of the results. They could also cite other experimental work that supports their argument that the critical FGF signal is mesenchymal, such as the work of Lauthier (1985), who showed that newt limbs can develop even when the apical epithelium is removed from the mesenchyme.

5) Figure 1C. Looking at the alcian blue stained limbs, there are no visible digit condensations at stage 47, and at stage 48, three distinct digit condensations are stained. At stage 49, these 3 condensations remain visible (only I and II are labeled in the figure, but digit III is also stained). The initial condensation of digit IV is also visible at stage 49. Given that no digits are visible at stage 47, and three digits (I, II, III) appear at stage 48, what is the basis for the argument that digits form from anterior to posterior? (see the following point for continued discussion of this issue)

6) Figure 1E. I found the *Sox9* in situ hybridizations to be less informative than the alcian blue preparations, possibly due to the level of background staining, although this could also be a result of slight staging differences between the alcian blue and *Sox9* limbs. For example, although one can see three digits in the alcian blue limb at stage 48, only digit II is obvious in the *Sox9* in situ hybridization at the same stage. Is the latter a slightly younger limb, or is the in situ just not as clear as the alcian blue preparation?

[Editors' note: further revisions were requested prior to acceptance, as described below.]

Thank you for resubmitting your work entitled "FGF-signaling is compartmentalized within the mesenchyme and controls proliferation during salamander limb development" for further consideration at *eLife*. Your revised article has been favorably evaluated by Marianne Bronner (Senior Editor), a Reviewing Editor, and two reviewers.

The manuscript has been improved but there are some remaining issues that need to be addressed before acceptance, as outlined below:

Essential revisions:

1) Please break the Discussion section into subsections, including one that compares the findings in axolotl with chick and mouse.

2) Please add a table that compares limb development in axolotl with mouse/chick.

3) Please try, if possible, within a reasonable time frame, to address the reviewer comments below, though some were deemed non-essential (e.g. scRNAseq of limb ectoderm; re-imaging the immunofluorescence). If you cannot address certain comments, please explain why.

*Reviewer #1:*

In this revised submission, Purushothaman et al., compare limb development in the axolotl salamander to classical findings in chicken and mouse embryos. Some findings confirmed previous studies using more refined techniques, other findings were new. Most striking were the numerous differences seen in axolotl including pre-axial condensation, condensation first along the metapterygial axis, lack of a morphological AER, lack of AER-FGF expression (instead found at the distal mesenchyme), no requirement for FGF signaling for Prox./Distal patterning or cell death, anterior expression of Shh, requirement of Shh for distal proliferation but not cell death. These observations are carefully presented and clearly discussed and very interesting, however, Although the authors responded to many of the criticisms in the first round with additional experiments, I am conflicted as to whether this is a collection of interesting descriptive phenomena vs. a comprehensive mechanistic conceptual advance in the field.

Essential revisions:

1) It would be nice to see some RNA-ISH that is positive in the overlying ectoderm just to rule out the possibility that lack of staining is not due simply to a technical problem with doing RNA-ISH on axolotl ectoderm (some permeability problem). Another method would be to do RNA-ISH on a section.

2) Please show Fgf4 RNA-ISH and other 'data not shown' such as cell death assays even if not very high.

3) How come only scSeq in mesenchyme is shown? Could an analysis of the ectoderm be included? In no FGF ligands detected in this data set, this would be nice to report.

4) Lysotracker and TUNEL are coincident in mouse/chick. Are they in axolotol? Could cells be dying but this is not visible due to failure of phagocytic bodies?

5) The second paragraph of the Discussion section does a better job at summarizing the paper than Abstract, perhaps the abstract could be enhanced with some of the ideas here.

6) Even for the limb bud aficionado it would be useful to have a chart comparing mouse/chick results to findings in axolotl.

7) In Figure 5H and Figure 6—figure supplement 1 panel F, I can't tell which of the red patches are positive above background. Also, in the mid-sagittal slice the control limb bud shows almost all cells as positive, is this correct?

8) Why are Christensen et al., 2002; Han et al., 2001 incorrect in their observation of FGF expression in the AER?

*Reviewer #2:*

The authors are to be congratulated for the quality and scope of the revised manuscript. It is a rigorous and extremely thorough study. All of my concerns have been addressed satisfactorily.

The DiI results are quite interesting. They injected DiI into the early Shh domain of the limb bud and then let it grow, examining the localization of labeled cells at different stages. This allowed them to map the spatiotemporal movements of the initial Shh domain and to determine whether the putative ZPA cells end up in the digits (and if so, which digits). This was very informative. Relative to chick and mouse, axolotl limbs have somewhat crazy fingers and gene expression patterns, but this experiment identifies a deeply conserved feature – the contribution of Shh-expressing cells to the posterior digit(s).

Expansion of the FGF analysis to include all of the Fgfs expressed in the AER of amniotes (FGF 4, 8, 9, 17) as well as Fgf10 and *Fgfr1* and *Fgfr2*, provides definitive evidence that the "AER Fgfs" of amniotes are expressed in the mesoderm of axolotl limb buds (with the possible exception of Fgf4, which was variable but never in the ectoderm). This is an unequivocal demonstration that axolotl limb buds lack the ectodermal domains of Fgfs that are found in the AER of amniotes.

The authors note that the proximal-distal segregation of FGF ligand and receptor expression is puzzling, but the results are clear. Moreover, the in situ hybridization patterns are validated by the scRNAseq data. It is also interesting to see that where *Fgfr1* does extend into the distal domain, it is either in or near the cells expressing FGF ligands.

I am impressed by the manner in which the authors responded to the reviewers' criticism that some conclusions were overly broad in the previous version of the paper. Rather than just removing conclusions that were not entirely supported by the results, they investigated whether those conclusions would stand up to further testing. In some places, they were just a half step away from having data that would address the question, but in other cases, they went to great lengths to carry out the appropriate experiments. Where the results provided clear support for the hypothesis (e.g., the AER Fgfs are expressed in the mesenchyme of axolotls), they say so, and where more than one interpretation is possible, they provide conservative conclusions and acknowledge the open questions that remain. I very much like this version of the paper and think that it will be a valuable contribution of the fields of limb development and limb evo-devo, and it will provide important developmental context for regenerative studies.

---

## [Author Response]

[…] We appreciate that addressing the points above is a significant amount of work. If you choose to address these and the other comments raised below and submit a more in-depth manuscript later on, we would be happy to re-examine the work. If, on the other hand, you would prefer to have the work published elsewhere as is we are enclosing the full text of the reviews, which we hope you will find useful. As you will see, both reviewers concurred on many aspects.

We appreciate the thorough reviews prepared by both reviewers and the summary provided by the reviewing editor. We have worked hard to address every point raised by the reviewers and address each of these points individual. Our revised manuscript contains a much more thorough analysis of the requested FGF ligands and receptors using WISH and scRNA-seq. We have included a more in-depth analysis of our inhibition experiments and now provide data on cell death and cell proliferation. We also use qRT-PCR to quantify gene targets of FGF- and Shh-signaling and provide an additional early time point. We have also reanalyzed our analysis of digit specification and differentiation. Lastly, we have reorganized our paper into a traditional format at the suggestion of R2 so as to provide a separate Discussion section at the end of the paper.

To facilitate your evaluation of the enclosed reviews, we have identified those comments which are essentially the same. I will use R1 and R2 to identify the reviewers, followed by point numbers of the respective reviews.

Using the similar points stated here, we refer to the comment below in response to R1 and then indicate in response to R2 that we have responded to the R1 comment.

R1-1 = R2-9;

Refers to AER structure.

We have corrected our language in some cases and include reference to direct-developing frogs and marsupials. Our data confirms that there is no MORPHOLOGICAL AER. Our new data covering the additional FGF ligands and receptors demonstrates there is not MOLECULAR AER.

R1-2 = R2-5 and R2-6;

Digit patterning and specification.

We agree with the reviewers and have completed a more detailed analysis using *Sox9* and Alcian blue staining. We present a deeper analysis as a Figure 1—figure supplement 2.

R1-4 = R2-4 and part of R2-3;

FGF ligand and receptors,

We have addressed these comments using in situ hybridization and scRNAseq analysis. New data is presented in Figure 3 (*FGF* expression), Figure 4, Figure 4—figure supplement 1 and Figure 4—figure supplement 2 (scRNAseq data).

Questions about the SU5402 treatments are discussed in R15, R2-3 and R2-14;

We now address these in the text and methods. We performed dose response which yielded the concentration we used. We administered during limb field specification prior to limb budding and thus believe our results accurately reflect inhibiting FGF-signaling from the outset of limb development. New data is also presented in Figure 5—figure supplement 1.

Different but complementary questions about Shh expression are raised in R1-6 and R2-1;

Our analysis of Shh expression and signaling now includes an analysis of cell death and cell proliferation and we have added a DiL labeling experiment at the request of R2. We present this data in Figure 2 and Figure 6.

R1-7 = R2-2

To address these comments we performed a cell death analysis during normal development and in response to FGF and Shh inhibition. We do the same for cell proliferation and use qRT-PCRT to quantify changes in gene targets following inhibition. This new data is presented in Figure 5, Figure 5—figure supplement 2 and Figure 6.

Reviewer #1:

This study makes the bold claim that Urodeles, as exemplified by the axolotl Ambystoma mexicanum, exhibit distinct developmental features compared to other tetrapod species. An intriguing implication of this claim is that a difference in developmental program may be related to the ability of axolotls to regenerate limbs. The study covers a wide variety of features: (1) The morphology of the limb bud, including whether or not an AER forms, (2) the order of deposition of an Alcian blue positive matrix and the expression of Sox 9, Ihh; (3) order of digit formation, (4) Proximal distal patterning (via Fgf8 expression, SU5402 treatment) (5) A/P specification (via Shh expression, cyclopamine treatment) (6) Relationship between the P/D and A/P regulatory loops (via Grem. Ptch1 regulation), and (7) D/V patterning (via Wnt, En1 expression etc).Unfortunately, the study handles each of these issues very superficially considering that there are at least 100 major studies examining each of these topics using chicken or mouse model systems over the course of the last 25 years. As a consequence, this reviewer believes that the bold conclusions drawn based on the data collected so far, while intriguing, are much too premature.

We fully appreciate all the detailed studies that have been done in chicken and mouse examining each of these issues during limb development. In light of this general comment, and the more detailed comments outlined below, we have expanded our analysis of FGF- and Shh-signaling in much greater detail and have reserved the D/V information for an additional paper.

Essential revisions:1) Just because there is no distal pseudo-stratified epithelial thickening (AER) doesn't mean an ectodermal signaling center is not there. There are other examples of amphibians that have no AER and still have AER-FGF expression.

We agree with the reviewer on this point. In the original submission we refer to the cases of *Eleutherodactylus coqui* (direct-developing frog) and *Monodelphis domestica* (opossum), both of which lack a morphological AER but demonstrate ectodermal *FGF* expression. In response to the comment, we have now been more careful with our language and clearly link *FGF*-expression where we are discussing the AER and highlight the two examples of the above-mentioned animals not having an AER, but still possessing an ectodermal signaling center (subsection “Amniote and anuran AER-specific *FGF* ligands (*8, 9, 17*) are expressed exclusively in axolotl limb mesenchyme” and Discussion section). We also believe data showing expression for FGF ligands and receptors further addresses this point.

2) Typically, condensation is defined by histological coalescing of cells, however this study does not use histology to determine the order of digit condensation. If condensation is defined by proxies for cartilage differentiation the authors need to explain the variation in Alcian blue intensity for digits (order = 2?1? or vice versa, not clear from images 3, 4), sox 9 (2,1, 3, 4), and Ihh (1, 2? 3, 4). The authors should compare their more careful findings with those of Sue Mackem and colleagues (PMID: 18410737 and 21509901). Axolotls have 4 digits, is it digit 1 that is missing or digit 5?

In response to the reviewer, we attempted to further analyze coalescing cells in developing axolotl limb buds using paraffin sections and semi-thin resin embedded sections but were unable to produce sections that could resolve the issues raised by in situ and Alcian blue samples. Instead, we revisited each and every limb stained for *Sox9* which we now present to the reader in a new supplementary figure (Figure 1—figure supplement 2). This data shows (pretty clearly in our opinion) that *Sox9* expression specifies digit II before digit I which appears shortly afterwards. In fact, *Sox9* expression can be seen emerging in the area of digit I in late stage 48 limb buds. The two digits are clearly stained at stage 49 for reference. In contrast, revisiting all of our Alcian blue stained limbs across these and later stages, we were unable to find an instance where digit II was differentiating before digit I. Therefore, we now present this data and draw the conclusion that the digits are specified in a slightly different temporal sequence than when they differentiate. Said another way, digits I and II differentiate at the same time even though digit I is specified before digit II. Rather than compare to the two Susan Mackem papers (which we are familiar with), we discuss how our findings support ontogenetic and heterchronic shifts that have occurred during urodele limb evolution. There is precedent for variation in the pattern of digit differentiation across urodeles, although no one has looked across many species using *Sox9*. With respect to the final point in this comment regarding digit identity, it is not our intention to get drawn into the controversy of digit identity, a debate which continues and is likely unresolvable without molecular markers for individual digits (see Towers et al., 2011). Thus, we have chosen not to address this in our manuscript.

3) The authors should be aware that colormetric RNA in situ hybridization is not as sensitive as other RNA detection methods (FISH or RNAscope). Thus, lack of color deposition, especially using whole mount preps, does not necessarily mean that no expression is present. For example, it may be the case that there is an early distal swath of Sox9 expression but that it present at too low levels to detect give the method used.

We appreciate that FISH and RNAscope have increased sensitivity compared to WISH. We also note that these methods have increased noise that creates problems when trying to demonstrate the absence of signal. With respect to the *Sox9* result, while our new supplementary figure presenting all *Sox9*-stained limbs for stages 47-49 cannot rule out a faint swath of expression in digit I as it is being expressed in digit II, we would expect that the same faint signal would be present prior to our observation of *Sox9* in digit II at stage 47. Thus, we do not believe that implementing an additional (and very costly) method would bring appreciable clarity to this point. We have been careful not to overstate our result which we believe should address this comment.

4) There are at least 22 Fgfs and 4 FGF receptors. In many comparisons between species, related growth factors have been shown to substitute for each other. For example, Sox10 appears to provide the function of mouse Sox9 in zebrafish. The expression pattern for Fgf8 shown in this study is reminiscent of the pattern of Fgf10 expression in chicken/mouse embryos. Without examining at least a handful of the most well-known FGF genes (Fgf4, Fgf10, Fgf9, Fgf2, Fgfr1, Fgfr2), it is premature to claim that axolotl limbs develop without needing an ectodermal source of FGFs.

As noted by the other reviewer we agree with this point. We now include WISH staining and scRNA-seq data for all the requested FGF ligands and receptors. Our data shows quite conclusively that none of the ligands or receptors are expressed in the ectoderm. These data also provide insight into segregation across the proximodistal and anteroposterior axes. We were unable to detect Fgf4 using in situ or in the scRNA-seq dataset.

5) SU5402 treatments typically result in a mild phenotype but based on their table of results CAN result in more severe digit loss phenotypes. This suggests that either the dose or timing of administration can be modulated to achieve phenotypes that match mouse KO phenotypes.(Perhaps best compared to Fgfr1/2 combo KOs). In addition, given the long period between establishment of the limb field and limb outgrowth, it may be that one would need to administer SU5402 much earlier (prior to limb bud outgrowth) to inhibit proximal specification in these species.

We appreciate the point raised here which is a result reminiscent of so many chick limb studies using pharmacological agents. Although a result can be robust, sometimes unseen effects to animal health or growth can have an additive effect on the ultimate phenotype. We administered SU5402 based on a dose-response study and selected the max dose that was not toxic and did not appreciable effect total growth of the animals. We administered SU5402 at stage 39 which is prior to limb bud outgrowth when the limb bud is not visible in an effort to inhibit FGF-signaling as soon as it began. Lastly, even in the 4/34 cases where additional elements were lost (“more severe phenotypes as referred to in the comment”), the missing elements were lost in the same posterior to anterior direction, rather than the proximal truncation predicted from AER loss of function studies in chicken and mouse. Furthermore, our new data showing that FGF-signaling regulates cell proliferation throughout the limb provides additional supporting evidence for our SU5042 results. Thus, our results are perfectly consistent with results from salamanders, anurans and chickens that show digit loss as a consequence of inhibiting cell proliferation (Discussion section). In our interpretation, loss of additional elements in a few (4/34) cases likely resulted from the small additive effect of a reduction in whole animal growth that was independent of FGF-signal inhibition.

6) The observation that Shh is also expressed in an anterior domain is interesting and should be examined. Given that cyclopamine administration gives the expected results, is this anterior domain not significant? Does the location of Shh expression correlate with areas of cell death at the anterior and posterior autopod margins?

We too find the anterior domain of *Shh* expression intriguing which is why we were careful to look for additional anterior expression of *Ptch1* and *Gli1*. These support that region as real expression. However, they do not speak to function. Therefore, at the suggestion of the reviewer, we used Lysotracker to analyze cell death throughout limb development. We did not, however, detect cell death in either the posterior or anterior domains. Likewise, our cell proliferation data does not show anything that specifically correlates with the anterior domain. In light of the cyclopamine results, we cannot at present assign a specific function to the anterior domain. It is feasible that the expression contributes to limb growth, but that when Shh-signaling is inhibited the effect is swamped out by the more serious effect on AP patterning and specification of skeletal progenitors.

7) The authors make the major claim that the FGF and Hh regulatory loops are not interlinked. This is based on examining a few read-outs by whole mount RNA in situ hybridization and is not comprehensively or quantitatively assessed. Furthermore, analysis of cell death is not presented, nor are genes or proliferation rates examined PRIOR to differences in limb bud size being evident.

We agree that our initial assessment could benefit from a deeper analysis. We now present additional evidence to determine if an FGF-Shh regulatory loop exists in salamander limbs. First, the reviewer makes a good point that we should have analyzed the limb buds for altered gene expression *prior* to differences in limb bud size. Thus, as the limb buds emerge from the flank at stage 44, we now present an analysis of stage 45 limb buds (Figure 5C,F and Figure 6D). In response to FGF inhibition (using SU5042) we show that *Etv1, Etv4* and *Gremlin1* expression are not present. In contrast, we show that *Ptch1* expression persists in a domain almost identical to the control limb. This data mimics our results at stage 46 (when the limb size is reduced). In addition to WISH, we now quantify gene expression changes at stage 46 and show that *Etv1, Etv4, Gremlin1* and *Shh* expression are significantly down-regulated. Again, our qRT-PCR data shows that *Ptch1* expression does not decrease (and in fact is trending towards a significant increase). This shows that Shh-signaling is not regulated by FGF-signaling. We present the same type of analysis using cyclopamine to inhibit Shh-signaling (Figure 6A-E). Our qualitative (WISH) and quantitative (qRT-PCR) analysis shows that *Fgf8* expression is regulated by Shh-signaling, as is *Gremlin1* expression. In addition to expression analysis, we now include functional assessment of cell death and cell proliferation. Cell death did not change in response to either treatment. Our cell proliferation analysis, however, showed that FGF-signaling regulated this process throughout the limb. This demonstrates that in contrast to anurans and amniotes where FGF-signaling controls cell survival and differentiation, in urodeles FGF-signaling controls cell proliferation instead. We obtained similar data for cyclopamine accept that Shh-signaling appeared to regulate cell proliferation to a lesser extent than FGF-signaling and that this control was more concentrated in the distal mesenchyme where *Fgf8* is expressed. In addition to presenting new data, we also discuss the implications for our findings in an expanded Discussion section.

8) Finally, the analysis of D/V patterning genes seems tacked on and could be eliminated from the paper.

We agree with the reviewer and have removed this data from the paper.

In its current state this reviewer recommends rejection of the paper as a thorough examination of the above issues would certainly take longer than typically allotted for eLife submissions. Should the bold claims hold after more detailed analysis, (suggest focusing on FGF and HH pathways as most interesting and critical, while describing the order of condensation/differentiation is likely less revelatory), the study could be of great interest and significance.

We believe the additional data and discussion addresses all of the reviewer’s comments and agree that the study will be an important contribution to the field.

Reviewer #2:

Summary:[…] They conclude that the mesenchymal-epithelial (Shh-Grem-Fgf8) feedback loop that controls limb development in other tetrapods has been re-arranged in axolotls, where it is entirely mesenchymal and has evolved different regulatory interactions. In a rich but short discussion, the authors explore the implications of these findings for understanding genetic interactions in the limb, evolution of developmental mechanisms, and molecular control of regeneration. The brevity of the discussion does a disservice to the many sacred cows that this paper has toppled, and I would encourage them to break apart the results and discussion so that a proper airing of these ideas can be presented. Overall, the data are clear and convincing and the discoveries are likely to be of interest to a broad group of biologists in the areas of development, regeneration and evolution.

We would like to thank R2 for their thorough assessment of our paper. As indicated by the Reviewing editor, several of the points raised below were identical to those raised by R1. Thus, where applicable, we note our response above and provide additional comment when necessary.

Essential revisions:1) The Shh expression pattern is quite interesting and makes one whether the cells that form the digits ever see the Shh signal. Although the initial domain in the posterior part of the limb is typical of early limb buds of tetrapods and fishes, the subsequent domains are very different from the patterns reported for other tetrapods. They state, "Posterior expression is never detected in the autopod" (Figure 2 legend). If the digits progenitors arise from (or near) the Shh domain, then the role of Shh in digit development could be conserved. If the cells of the Shh domain are distant from the digit progenitors, then this would challenge the current paradigm of digit development. This could be tested quite easily by labeling the initial Shh domain (at stage 45) with an injection (or carefully applied crystals) of an indocarbocyanine dye, like DiI, and then examining the distribution of labeled cells several stages later in order to determine whether those cells contribute to the digits. Determining whether digit progenitors reside within the Shh domain would allow the authors to make a direct comparison with the Shh-expressing (ZPA) cells of mouse and chick limb buds, which contribute to posterior digits.

We have responded above to the comment about the anterior domain of *Shh* in comment reviewer 1, comment 6. We present data in a series of in situs (summarized in a schematic) showing the posterior expression domain of *Shh* relative to similar stages in chick and mouse (Figure 2C). This shows that in salamanders, *Shh* expression remains outside of the autopod after limb bud elongation. While the functional significance of this extended expression remains unclear, we appreciate the reviewer’s suggestion to use DiL as a way to investigate conservation of the posterior domain with respect to establishing digit progenitors. We performed the suggested experiment by injecting DiL at stage 45 when *Shh* is first expressed in the posterior and then tracked the labeled cells in limbs until all the digits had formed at stage 53. We present this data in a new figure alongside expression for *Shh, Ptch1, Gli1* and *Gli3* (Figure 2). Our results mirror similar experiments in chick limbs showing that the region expressing *Shh* gives rise to the most posterior digit and that later Shh-signaling controls growth of the autopod. When viewed alongside our data using cyclopamine (and that previously published by Stopper and Wagner, 2007), it appears Shh-signaling controls digit progenitor specification and later distal cell proliferation. Similar to mouse and chick, digit 1 specification in salamanders is not regulated by Shh-signaling.

2) Because the SU5402 treatments show that FGF signaling does not appear to maintain Shh in axolotls, is there really is a FGF-Shh feedback loop in axolotl limb bud mesenchyme?

This is a good point. We have revised our manuscript accordingly. We still refer to the feedback loop as it applies to anurans and tetrapods and where we refer to the interaction of Shh and FGF in salamanders indicate that it is not a loop.

3) Do the authors think that the same mesenchyme cells that are expressing Fgf8 are also responding to Fgf8? Do they express an appropriate receptor? How does the topological shift in the position of the ligand, Fgf8, relate to the ability of SU5402, a FGF receptor antagonist, to produce such different outcomes in axolotl vs chick and mouse limbs? Do the authors think that there is combinatorial FGF signaling in axolotls (as there is in chick and mouse), but the ligands are produced and received by the mesenchyme? If so, then can they suggest an explanation for how the same Fgfs could regulate Shh when secreted by the AER(as in chick/mouse) but do not regulate Shh when produced by the mesenchyme? Chick experiments in which the AER is removed and FGF beads are implanted into the limb bud mesenchyme suggests that the source of the ligand is irrelevant. Have they considered the possibility that Fgf8 acquired a novel function during axolotl limb evolution and lost its roles as a Shh regulator and an outgrowth signal? The proliferation data suggest that at least some of these roles are conserved.

The reviewer raises good points with respect to signal source and receiver. In an attempt to clarify what is happening during salamander limb development with respect to FGF-signaling, we took two additional approaches. First, we used in situ hybridization to detect additional FGF ligands that act as AER signals in chick and mouse (i.e., *Fgf4, 9* and *17*) and the two primary FGF receptors (i.e., *Fgfr1* and *2*). We also looked at *Fgf10* since it is normally expressed in the mesenchyme, but signals to *FgfR2iiib* expressed in the ectoderm. We now present all of this data in a new Figure 3. Two things are apparent from this data: (1) all of the ligands (except *Fgf4* which we could not detect) are expressed in the distal mesenchyme and none are expressed in the ectoderm and (2) the cognate receptors are expressed mostly in proximal cells that do not appear to overlap with the ligands. One exception is *Fgfr1* which appears to be expressed more broadly at stages 44 and 45, although even then it is restricted from the most distal mesenchyme. Second, we tried to address localization of FGF-signaling in the limb and whether the same mesenchymal cells that are expressing *FGF* ligands are also responding to FGF-signaling. To do this we analyzed a single cell RNA-seq (scRNAseq) dataset generated for two early stages of axolotl limb development that corresponded to our stages 44 and 45 (Gerber et al., 2019). We present the results from this analysis in a new Figure 4, and two supplementary figures. Using this dataset, we conducted a principal component analysis of the sequenced cells and used known genetic markers of the proximodistal axis to discover if a principal component could accurately model the proximodistal axis. This analysis revealed a proximodistal axis as defined by expression of *Meis1* and *Meis2* (proximal markers) and *Hoxd11* (distal) (presented in Figure 4—figure supplement 1). Next, we conducted a co-expression analysis which showed that the FGF ligands and receptors are in fact, not co-expressed along the modeled proximodistal axis. This data is now presented alongside the in situs in Figure 4. We also asked whether *Fgf8, 9, 10* and *17* were co-expressed in cells that also expressed any of the four *FGF* receptors. This analysis is presented in Figure 4—figure supplement 2 and shows that some cells expressing *Fgfr1* also express these four ligands. However, few to no cells co-express the ligands and other receptors. Together, we believe these approaches present compelling evidence that distal mesenchyme cells expressing ligands are signaling to more proximally expressed receptors, but that some distal mesenchymal cells may be able to respond in an autocrine fashion.

With respect to the second point regarding an explanation for breaking the Shh-FGF feedback loop; as discussed above, Shh-signaling appears to maintain a conserved function with regard to anteroposterior patterning, specification of digit progenitor cells, and distal growth of the autopod. Presumably, movement of FGF-signaling to the mesenchyme occurred after the divergence of urodeles and anurans and our cell proliferation and cell death data suggest that FGF-signaling adopted a novel function in supporting growth of the limb rather than controlling cell survival and proximodistal pattern. We discuss this in relation to results from salamanders, anurans, and amniotes which show similar phenotypes when cell proliferation is broadly inhibited. How this happened in the context of evolution is much more difficult to answer and we are not sure it can be, at least not in the scope of this manuscript.

4) In several places, the Fgf8 result is generalized to the other Fgfs involved in AER signaling (e.g., Results and Discussion section: "Taken together, our expression data clearly shows that the Shh-Grem-FGF feedback loop is entirely expressed within the mesenchymal compartment during salamander limb development."). In chick and mouse limbs, at least 4 Fgfs (Fgf4, Fgf8, Fgf9, and Fgf17) are expressed in the AER and signal through mesenchymal FGF receptors. Whilst it would be informative to examine the expression patterns of these other Fgfs in axolotl limb buds to determine whether the Shh-FGF feedback loop is entirely in the mesenchyme (or if some of the Fgfs have remained in the epithelium) if they choose not to examine the expression of these other Fgfs, then they should tailor the discussion accordingly and at least mention the alternate possibilities. For example, they show that Fgf8 is expressed in the mesenchyme, but it is unknown whether or where other Fgfs are expressed in the limb. Thus, several scenarios are possible, including (a) Fgf8 is mesenchymal but Fgf4, Fgf9, and Fgf17 are expressed in the epithelium, (b) All 4 Fgfs are mesenchymal, (c) Fgf8 is mesenchymal and the other 3 Fgfs are not expressed. Without knowing which of these scenarios is true, one must be cautious and clear with the interpretation the SU5402 results since SU5402 treatment probably suppresses all FGF signaling. For example, scenarios (b) and (c) would support "a topological rearrangement of the Shh-Grem1-FGF loop", but scenario (a) would not because 3 of the 4 Fgfs would be expressed in ectoderm. To be clear, I am not insisting that the authors examine every FGF and FGF receptor in axolotl limbs, but they need to consider the complexity of FGF signaling in amniotes in their interpretation of the results. They could also cite other experimental work that supports their argument that the critical FGF signal is mesenchymal, such as the work of Lauthier (1985), who showed that newt limbs can develop even when the apical epithelium is removed from the mesenchyme.

We direct the reviewer to our responses reviewer 1, comment 4 and reviewer 1, comment 5 above.

5) Figure 1C. Looking at the alcian blue stained limbs, there are no visible digit condensations at stage 47, and at stage 48, three distinct digit condensations are stained. At stage 49, these 3 condensations remain visible (only I and II are labeled in the figure, but digit III is also stained). The initial condensation of digit IV is also visible at stage 49. Given that no digits are visible at stage 47, and three digits (I, II, III) appear at stage 48, what is the basis for the argument that digits form from anterior to posterior? (see the following point for continued discussion of this issue)6) Figure 1E. I found the Sox9 in situ hybridizations to be less informative than the alcian blue preparations, possibly due to the level of background staining, although this could also be a result of slight staging differences between the alcian blue and Sox9 limbs. For example, although one can see three digits in the alcian blue limb at stage 48, only digit II is obvious in the Sox9 in situ hybridization at the same stage. Is the latter a slightly younger limb, or is the in situ just not as clear as the alcian blue preparation?

We direct the reviewer to our response to reviewer 1, comment 2

[Editors' note: further revisions were requested prior to acceptance, as described below.]

Essential revisions:1) Please break the Discussion section into subsections, including one that compares the findings in axolotl with chick and mouse.2) Please add a table that compares limb development in axolotl with mouse/chick.3) Please try, if possible, within a reasonable time frame, to address the reviewer comments below, though some were deemed non-essential (e.g. scRNAseq of limb ectoderm; re-imaging the immunofluorescence). If you cannot address certain comments, please explain why.

Following the essential revisions provided, we have broken up the Discussion section into subheadings and specifically included one where our findings in axolotl are compared with *Xenopus*, chick and mouse. In this same section we now reference a new Table 1 detailing these comparisons. In response to the specific comments from the two reviewers we have outlined our point-by-points replies below.

Reviewer #1:

[…] Essential revisions:1) It would be nice to see some RNA-ISH that is positive in the overlying ectoderm just to rule out the possibility that lack of staining is not due simply to a technical problem with doing RNA-ISH on axolotl ectoderm (some permeability problem). Another method would be to do RNA-ISH on a section.

We understand the request for this piece of data. To demonstrate the reliability of our protocol at detecting ectodermal, as well as mesenchymal expression in developing axolotl limb buds, we now include an additional supplementary figure (Figure 2—figure supplement 1). This figure shows Msx2 staining (which is absent from the ectoderm) in the distal mesenchyme alongside in situs for two Keratin genes, Krt5 and Krt17. Staining for these genes is observed broadly across the surface ectoderm. To further show these genes are expressed in the ectoderm, we provide sections of our embryos where localization is clearly demonstrated in the overlying ectoderm (and is not expressed in mesenchymal cells).

2) Please show Fgf4 RNA-ISH and other 'data not shown' such as cell death assays even if not very high.

We appreciate the request to include these data which we now include as additional figure supplements. We now present limb buds negative for Fgf4 expression in Figure 3—figure supplement 1 which show no expression at stages 44-46. We have expanded Figure 5—figure supplement 2 to show the lack of cell death during normal development at stage 44 and 46 using Lysotracker Red. We also now include data for stage 46 limb buds from control and treatment animals from our SU5402 and cyclopamine experiments.

3) How come only scSeq in mesenchyme is shown? Could an analysis of the ectoderm be included? In no FGF ligands detected in this data set, this would be nice to report.

The scRNAseq data we analyzed and that was reported in Gerber et al., (2018) for limb buds excluded ectodermal cells. The authors of that paper did not release the data for the ectodermal cells they excluded. We had originally hoped to do this, but discovered they had excluded these cells.

4) Lysotracker and TUNEL are coincident in mouse/chick. Are they in axolotol? Could cells be dying but this is not visible due to failure of phagocytic bodies?5) The second paragraph of the Discussion section does a better job at summarizing the paper than Abstract, perhaps the abstract could be enhanced with some of the ideas here.

We agree and we originally had an abstract that better reflected this summary in the Discussion section.

However, *eLife* only allows an abstract of 150 words and this length does not allow much detail. Thus, the current Abstract attempts to present several key findings instead of every one we detail in the paper.

6) Even for the limb bud aficionado it would be useful to have a chart comparing mouse/chick results to findings in axolotl.

We now include a table (new Table 1) that appears with the discussion. The table compares salient features of forelimb development between axolotl, *Xenopus*, chick and mouse.

7) In Figure 5H and Figure 6—figure supplement 1 panel F, I can't tell which of the red patches are positive above background. Also, in the mid-sagittal slice the control limb bud shows almost all cells as positive, is this correct?

We believe the reviewer is referring to the lightsheet images depicting cell proliferation in Figure 5 and Figure 6 (so not the supplement for Figure 6). Provided this is the case, all the red “patches” are in fact, EdU+ cells. There is no background. Because the lightsheet provides a threedimensional projection of all the data, variations in brightness refer to cells that are closer to the front of the projection. As for the slice the reviewer is referring too (mid-longitudinal), yes, almost all cells are positive for EdU at this stage.

8) Why are Christensen et al., 2002; Han et al., 2001 incorrect in their observation of FGF expression in the AER?

First, we inadvertently cited Christensen et al., 2002 along with Han et al., 2001. Han et al., 2001 present whole mount expression data and state that “Fgf8 is expressed in limb bud epidermis and mesenchyme of the budding limb”. They go on to state in their discussion, “In the present study, we have found that the FGF-8 expression domain is translocated from epidermis to mesenchyme gradually as limb development proceeds”. We simply state that this is not what we found because at no time do we observer ectodermal expression of any FGF ligands. The quality of their images makes it impossible to determine in what compartments Fgf8 is or is not expressed. So all we are left with is what they state in their paper. Christensen et al., 2002 state, “Fgf8 and 10, highly expressed in the amniote developing limb AER and mesoderm, respectively (Crossley et al., 1996; Ohuchi et al., 1997), were accordingly also expressed in axolotl developing and regenerating limbs”. Thus, they imply that Fgf8 is expressed in axolotl limb ectoderm. However, they do not actually show an in situ for developing limbs in that paper. Therefore, it is inappropriate to cite Christensen et al., 2002 since they do not present data to support or refute our data. Thus, we have removed the Christensen et al., 2002 citation.